# Partial Observation Inversion and Batched Belief-State Planning for Information Gathering POMDPs

## Abstract

We present the *inversion variational autoencoder* ($\mathcal{I}$-VAE), a conditional generative model for efficient belief-state planning in partially observable sequential decision making problems. The $\mathcal{I}$-VAE maps partial observations to stochastic posterior state samples by learning an observation-conditioned latent prior, enabling consistent belief updates without an explicit likelihood model. We further fine-tune the belief model with a trajectory-based mutual information objective to improve latent space consistency across observation sequences. To support scalable planning with these learned beliefs, we formulate the *batched belief-state Markov decision process*, which is designed to parallelize rollouts while preserving optimality in expectation. We analyze heuristic policies that maximize the expected entropy reduction of the updated belief and show that these heuristics result in the optimal one-step expected Bayesian information gain. Our approach is evaluated on a benchmark masked-pixel task and a real-world intrusion discovery task using indirect muon tomography data, showing improved estimation accuracy and planning efficiency over conventional methods.

## 1 Introduction

In many real-world decision making problems, an agent must select an action at each timestep based on limited or uncertain knowledge of its current state. While the underlying environment may be modeled as a standard Markov decision process (MDP), effective planning under uncertainty often requires reasoning about multiple possible outcomes or future states. This motivates the use of *batched planning* methods, in which a set of sampled states are used to evaluate the expected performance of candidate actions. Such approaches have been widely adopted in belief-space planning (Kochenderfer et al., 2022), distributionally robust decision making (Nilim & El Ghaoui, 2005; Iyengar, 2005), and batched reinforcement learning algorithms (Osband et al., 2016; Lee et al., 2021), where parallel rollouts are used to compute uncertainty-aware value estimates. In this work, we formalize a batched extension of the MDP and partially observable MDP (POMDP) used for planning and address parallelized belief updating in high-dimensional observation spaces. The resulting batched model enables efficient, parallelized rollouts while ensuring that only a single action is selected and executed in the true environment. The batched planning formulation is motivated by the emphasis on modern GPU architectures for large-scale planning tasks, which can exploit the independence across batches to simulate transitions and compute rewards in parallel (Steinkraus et al., 2005).

To address parallelized belief updating in high-dimensional observation spaces, we study methods that sample from the posterior state distribution conditioned on a set of partial observations for information gathering POMDPs. We consider the case of the *purely epistemic Markov decision process* (EMDP) (Sabbadin et al., 2007), which is a special case of a POMDP where we do not have state transition dynamics, i.e., actions do not affect the (static) state. This type of problem can be seen as an information gathering problem where uncertainty in the hidden state needs to be reduced to a final decision. Examples of such problems include preference elicitation (Sabbadin et al., 2007) and critical mineral exploration (Mern & Caers, 2023). As a real-world test case, we evaluate our surrogate belief updater on a geological planning problem that inverts muon tomography data to a distribution of subsurface intrusion states. Muons are naturally occurring cosmic rays that have been used to probe internal structures of large objects. Muon tomography has been used in problems such as subsurface mineral exploration (Schouten & Ledru, 2018; Schouten, 2019) and

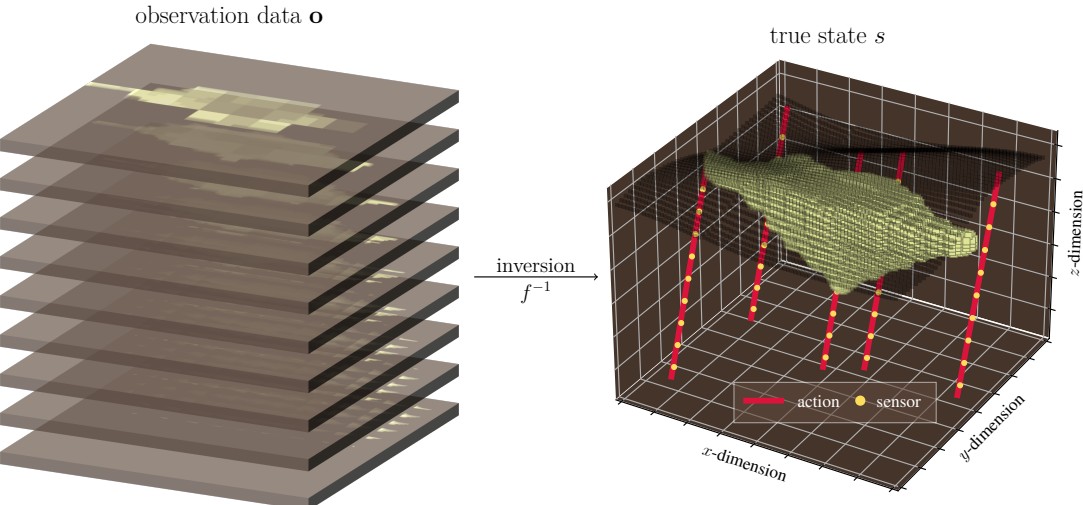

Figure 1: Inverting muon density data from 9 depth sensors to an estimated intrusion state.

understanding the internals of the Great Pyramids (Morishima et al., 2017), making this a challenging real-world information-gathering problem.

Using a prior dataset $(s^{(i)}, o_{1:t}^{(i)}) \in \mathcal{D}_{\text{partial}}$ over the states $s \in \mathcal{S}$ and observations $o_{1:t} \in \mathcal{O}$, surrogate models have been studied that approximate the posterior $p(s \mid o_{1:t})$ given a set of observations $o_{1:t}$ over time. Samples from this posterior act as samples from the updated belief. Because we are addressing EMDP problems, we can simplify the belief update in standard POMDPs. This simplification results in an *inversion* problem (Tarantola, 1987). Inversion is the process of going from a set of (partial) observations $o_{1:t}$ to the true state (or a distribution of states) as shown in Figure 1, denoted as $s = f^{-1}(o_{1:t}, a_{1:t})$ or $s \sim f^{-1}(o_{1:t}, a_{1:t})$ for stochastic processes. The name stems from the inverse of the *forward* observation model $o_{1:t} = f(s, a_{1:t})$ or $o_{1:t} \sim O(\cdot \mid a_{1:t}, s)$ in POMDP notation. Note that in an EMDP, the static state does not transition based on actions, hence $s' = s$. For brevity, we use the shorthand $\mathbf{o}$ instead of $o_{1:t}$ (indicating a vector of observations up to time $t$). To define our belief, we take $k$ samples from the posterior $p(s \mid \mathbf{o})$ to get a finite vector of posterior states $b = (s_1, s_2, \ldots, s_k)$ where $s_i \sim p(\cdot \mid \mathbf{o})$. Similar to particle filtering (Thrun et al., 2005), our belief is defined by a set of state particles $b$, yet we do not use the observation likelihood function $O(o \mid a, s)$. Instead, we only require observations generated from states for training (i.e., the forward model), without needing to evaluate their likelihood.

To develop a likelihood-free posterior sampling method for POMDP planning, we extend the *conditional variational autoencoder* (CVAE) model (Sohn et al., 2015) and fine-tune to maximize mutual information across observations from the same state—therefore, allowing latent observation trajectories to be more consistent across time. As with the CVAE, we train using states and partial observations but only require partial observations during inference for conditioning. Our work can be viewed as learning a generative model that samples states that are reconstructed from partial observations. For our work, we consider the states and observations to be images (partial images for observations), therefore, generative modeling research from the computer vision community can be directly applicable.

**Probabilistic generative models.** A variety of deep generative models have been proposed to learn complex data distributions and enable conditional sampling. Variational autoencoders (VAEs) (Kingma & Welling, 2013; Rezende et al., 2014) frame the density estimation problem as approximate Bayesian inference and train an encoder-decoder architecture to maximize a variational lower bound on the data likelihood $p_\theta(s)$. Conditional VAEs (CVAEs) extend this framework by incorporating conditional information (e.g., labels or partial observations) into both the encoder and decoder, enabling conditional generation via $p_\theta(s \mid \mathbf{o})$ (Sohn et al., 2015). We detail the VAE and CVAE generative model architectures in the following section. Other approaches, such as generative adversarial networks (GANs) (Goodfellow et al., 2014), formulate generation

as a two-player minimax game between a generator and a discriminator, achieving high-fidelity samples but often suffering from instability and mode collapse (Metz et al., 2017). Mirza & Osindero (2014) introduced conditional GANs (CGANs), which extend traditional GANs by guiding the generator-discriminator framework with conditioning variables. Beyond VAEs and GANs, normalizing flows (Rezende & Mohamed, 2015) provide exact likelihood evaluation through a sequence of invertible transformations. Score-based and diffusion models (Ho et al., 2020; Song et al., 2021) demonstrate state-of-the-art sample quality by learning to reverse a data corruption process, and have been applied to simulation-based posterior estimation through conditional score matching (Sharrock et al., 2024), joint distribution modeling with transformers (Gloeckler et al., 2024), and flow matching with continuous normalizing flows (Dax et al., 2023). However, while these models offer density evaluation by integrating the probability flow ODE, they require many iterative forward passes, making them prohibitively slow for the rapid belief updates and distributional reasoning required in POMDPs. Consistency methods (Song et al., 2023; Schmitt et al., 2024) reduce this sampling burden to as few as 1–4 forward passes but do so by sacrificing access to both the score and the density entirely, yielding only empirical samples. In settings that require explicit distributional reasoning over beliefs, such as evaluating information gain during planning, relying purely on sample-based estimators or expensive ODE integrations is limiting. We extend deep conditional generative models to support inversion problems by ensuring reconstruction consistency and guiding fine-tuning using mutual information to generate state samples given partial observations. Cao et al. (2023) provide an overview of different classes of generative models.

**Deterministic inversion methods.** In geological research, inversion problems (Tarantola, 1987) can be viewed as computing a function that reconstructs some hidden state of the world given some (noisy) observation, generally geophysical measurements such as gravity data (Huang et al., 2021), active seismic imaging (Deng et al., 2022), or density data from passive muon tomography (Lechmann et al., 2021; Schouten et al., 2022). Deep learning models have been applied to geological inversion problems primarily through the use of deterministic convolutional neural networks (CNNs) (Puzyrev, 2019; Hu et al., 2021). In these settings, a neural network is trained to perform regression $s = f_\theta(\mathbf{o})$ given full observations up to the horizon $T$. In our work, we consider the case where the observations are only partially observed (e.g., partial borehole data or partial geophysical data), yet a full state is sampled from the model. Other approaches, such as linear and nonlinear least-squares methods as well as regularization techniques, have been explored in the deterministic inversion literature (Menke, 1989; Tikhonov, 1963; Constable et al., 1987; Parker, 1994). Principal component analysis (PCA) and kernel PCA have also been used as dimensionality reduction techniques for deterministic history matching (Reynolds et al., 1996; Esmaeili et al., 2020). Tarantola & Valette (1985) and, more recently, Zhdanov (2015) provide comprehensive reviews of existing deterministic inversion methods applied to geological problems.

**Probabilistic inversion methods.** To quantify the uncertainty in the hidden state, probabilistic inversion methods—similar to probabilistic generative models—have been extensively studied. McAliley & Li (2024) use a straightforward implementation of a conditional VAE to invert geophysical data and sample from a unit Gaussian $z \sim \mathcal{N}(\mathbf{0}, \mathbf{I})$ during inference, as is standard in most CVAE applications. Their approach omits any observation encoder as the observations are vectors of synthetic gravity. In our work, we are interested in high-dimensional observations where a separate observation encoder is useful for dimensionality reduction and we learn latent Gaussian parameters $\mu_\mathbf{o}$ and $\log \sigma_\mathbf{o}^2$ directly. Chung et al. (2024) proposed a paired autoencoder approach but rely on an initial guess of the true state during inference and learn an observation decoder alongside a state decoder, which is not necessary for strictly observation-to-state inversion, as we show in our experiments. Other sampling techniques have also been studied such as Markov chain Monte Carlo (MCMC) (Metropolis et al., 1953; Hastings, 1970; Mosegaard & Tarantola, 1995). Laloy et al. (2017) condition based on expensive MCMC steps and suggest long runs to generate larger Markov chains for full exploration of the posterior, which may be expensive for planning. In a recent survey on neural network-based geological inversion, Li et al. (2023) identify a need for efficient probabilistic inversion methods.

**Parallel and batched online planning.** The use of batching and parallelism in online planning is well established, particularly in Monte Carlo tree search. Early work studied parallel MCTS variants based on tree parallelism, root parallelism, and leaf parallelism (Cazenave & Jouandeau, 2008; Chaslot et al., 2008; Bourki et al., 2010). More recent work has also exploited batching for neural-network-guided MCTS, where batched

GPU inference can substantially reduce evaluation cost (Cazenave, 2022). Other work considered more structured batched selection rules, including batch value of perfect information for MCTS (Shperberg et al., 2017), and parallel online POMDP planning algorithms such as HyP-DESPOT (Cai et al., 2021). Hoerger et al. (2025) further explored vectorized online POMDP planning. Our contribution is complementary: rather than proposing a new tree-search parallelization scheme, we formalize the batched belief-state transition induced by learned generative belief updaters and study how likelihood-free posterior sampling can be used inside information-gathering POMDPs.

## 2   Background on POMDPs

Sequential decision making under uncertainty is commonly formalized using Markov decision processes and their partially observable extensions. In information-gathering problems, the agent does not fully observe the true state of the world, but instead chooses actions that reveal partial, noisy, or indirect observations. We first review the partially observable Markov decision process (POMDP), then the purely epistemic reduction considered in this paper.

### 2.1   POMDP

The *partially observable Markov decision process* (POMDP) is a stochastic single-agent model for sequential decision making under partial observability. We define a POMDP as the tuple $\langle \mathcal{S}, \mathcal{A}, \mathcal{O}, T, R, O, \gamma \rangle$, where:

- $\mathcal{S}$ is the state space,

- $\mathcal{A}$ is the action space,

- $\mathcal{O}$ is the observation space,

- $T : S \times \mathcal{A} \to \Delta(\mathcal{S})$ is the state-transition kernel, where $T(s' \mid s, a)$ denotes the probability or density of transitioning to state $s'$ after executing action $a$ in state s,

- $R : \mathcal{S} \times \mathcal{A} \to \mathbb{R}$ is the reward function, where $R(s, a)$ is the immediate reward for executing action $a$ in state $s$,

- $O : \mathcal{A} \times \mathcal{S} \to \Delta(\mathcal{O})$ is the observation kernel, where $O(o \mid a, s')$ denotes the probability or density of observing $o$ after action $a$ results in state $s'$, and

- $\gamma \in [0, 1)$ is the discount factor.

Because the true state is not directly observed, the agent maintains a belief state $b \in \Delta(\mathcal{S})$, where $b(s)$ denotes the posterior probability or density assigned to state $s$. Given a current belief $b$, an action $a$, and a new observation $o$, the Bayesian belief update is:

$$b'(s') = \eta \, O(o \mid a, s') \int_S T(s' \mid s, a) b(s) \, ds \tag{1}$$

where $\eta$ is a normalizing constant. For finite state spaces, the integral is replaced by a sum:

$$b'(s') = \eta \, O(o \mid a, s') \sum_{s \in S} T(s' \mid s, a) b(s). \tag{2}$$

Transition kernel $T$ propagates the current belief through the dynamics, and observation kernel $O$ reweights the predicted belief according to the newly received observation. The belief-state formulation may be used to convert a POMDP into a Markov decision process over beliefs known as a Belief MDP. The belief reward is:

$$R_b(b, a) = \int_S b(s) R(s, a) \, ds \tag{3}$$

and a belief-state policy $\pi : \Delta(\mathcal{S}) \to A$ selects actions from the current belief. The planning objective is to find a policy maximizing expected discounted return:

$$\pi^* \leftarrow \arg\max_\pi \mathbb{E}_\pi \left[ \sum_{t=0}^{H-1} \gamma^t R_b(b^t, \pi(b^t)) \right] \tag{4}$$

where $H$ is the planning horizon. The finite sum is replaced by an infinite discounted sum in infinite-horizon settings.

## 2.2 Epistemic MDP

The *purely epistemic Markov decision process* (EMDP) is a special case of a POMDP with a hidden static state. Actions do not change the underlying state, and instead determine what information is observed. Formally, an EMDP can be written as a POMDP $\langle \mathcal{S}, \mathcal{A}, \mathcal{O}, T, R, O, \gamma \rangle$ with identity transition dynamics $T(s' \mid s, a) = \mathbf{1}\{s' = s\}$. Substituting the identity transition into the POMDP belief update gives:

$$b'(s) = \eta \, O(o \mid a, s) b(s). \tag{5}$$

The EMDP belief update does not propagate uncertainty through state dynamics and only incorporates new information about the fixed hidden state. This makes the EMDP belief update an inverse problem.

Let $s \in S$ denote the fixed but hidden state. A sequence of sensing actions $a_{1:t}$ produces partial observations $o_{1:t}$ through a forward observation process. In the deterministic case, we write $o_{1:t} = f(s, a_{1:t})$, equivalently $o_j = f(s, a_j)$ at each sensing step; in the stochastic case, observations are drawn according to $o_j \sim O(\cdot \mid a_j, s)$ for $j = 1, \ldots, t$. The corresponding inverse problem is to infer the hidden state, or a distribution over possible hidden states, from the accumulated partial observations, which we write as $s \sim f^{-1}(o_{1:t}, a_{1:t})$. If the likelihood $O(o \mid a, s)$ is available and the observations are conditionally independent given the hidden state and selected actions, then the posterior can be written as:

$$p(s \mid o_{1:t}, a_{1:t}) \propto b_0(s) \prod_{j=1}^{t} O(o_j \mid a_j, s) \tag{6}$$

where $b_0$ is the initial belief over states. However, in the likelihood-free setting considered in this work, the observation likelihood may be unavailable, expensive to evaluate, or only implicitly represented by a simulator or dataset. We therefore learn a conditional generative belief updater that maps partial observations to posterior state samples.

**Problem 1 (Likelihood-free planning in EMDPs)** *Let $\langle \mathcal{S}, \mathcal{A}, \mathcal{O}, T, R, O, \gamma \rangle$ be an EMDP with static hidden state $s \sim b_0$ and identity transitions. The action space decomposes as $\mathcal{A} = \mathcal{A}_{sense} \cup \mathcal{A}_{dec}$: sensing actions $a \in \mathcal{A}_{sense}$ incur cost $c(a) > 0$ and yield observations $o \sim O(\cdot \mid a, s)$, while terminal decisions $a \in \mathcal{A}_{dec}$ end the episode with reward $R_d(s, a)$. The likelihood $O(o \mid a, s)$ cannot be evaluated; only a dataset $\mathcal{D} = \{(s^{(i)}, o_{1:T}^{(i)})\}$ sampled from the forward model is available. The objective is to find a belief-state policy $\pi$ maximizing:*

$$J(\pi) = \mathbb{E}\left[ \sum_{t=0}^{\tau-1} \gamma^t \big(-c(a_t)\big) + \gamma^\tau R_d(s, a_\tau) \right] \tag{7}$$

*where $\tau = \min\{t : a_t \in \mathcal{A}_{dec}\}$ is the decision time induced by $\pi$, and the expectation is over $s \sim b_0$ and the observation process.*

Maximizing $J(\pi)$ poses two challenges: (i) maintaining the belief $b_t(s) = p(s \mid o_{1:t}, a_{1:t})$ without access to the likelihood $O$, and (ii) efficiently evaluating expectations over future observations and beliefs during planning. Section 4 addresses (i) with a learned generative belief updater, and Section 5 addresses (ii) with the batched belief-state MDP.

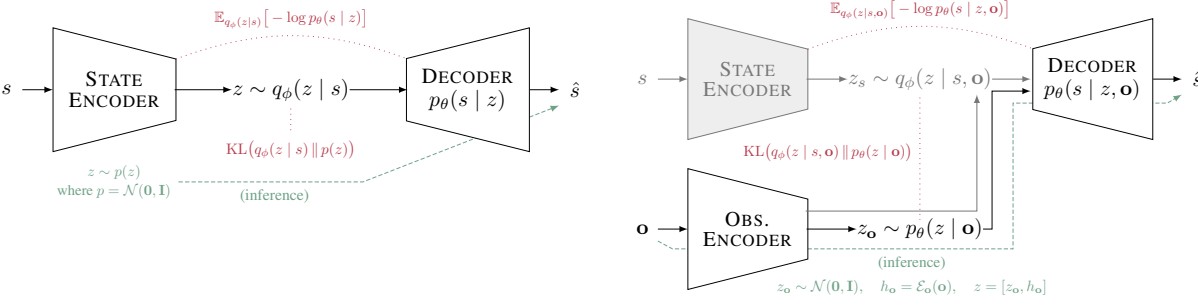

(a) The variational autoencoder (VAE).   (b) The conditional variational autoencoder (CVAE).

Figure 2: Traditional variational autoencoder architectures and losses.

# 3 Background on Variational Autoencoders

The batched MDP provides a scalable framework for belief-state planning, but it also requires parallelizable belief models to be effective. In partially observable settings, this means being able to generate batches of consistent posterior state samples from partial and noisy observations. Variational autoencoders (VAEs) and their conditional variants offer a natural foundation, since they provide efficient generative models for approximating complex posteriors. We first review VAEs and conditional VAEs, then introduce the inversion variational autoencoder ($\mathcal{I}$-VAE), which adapts these ideas specifically for planning.

## 3.1 Variational Autoencoder

Kingma & Welling (2013) introduced the *variational autoencoder* (VAE), a probabilistic model used for approximate inference with continuous latent variables. The VAE is designed to approximate the generative model $p(s)$ by assuming the variables $s$ are sampled following some hidden process. This process consists of first sampling a hidden random variable $z \sim p_\theta(z)$, then generating a state sample using the latent $z$ as a condition $s \sim p_\theta(s \mid z)$. This assumes both the prior $p_\theta(z)$ and likelihood $p_\theta(s \mid z)$ come from some parametric family of distributions (Kingma & Welling, 2013). To approximate the intractable true posterior density of $p_\theta(z \mid s) = p_\theta(s \mid z)p_\theta(z)/p_\theta(s)$, a probabilistic *encoder* model $q_\phi(z \mid s)$ is used (also called the *recognition model*). Similarly, $p_\theta(s \mid z)$ is referred to as the probabilistic *decoder* (or the *generation model*).

The objective when training a VAE is to maximize the marginal likelihood $p(s^{(i)})$. Kingma & Welling (2013) show that the objective can be approximated by the *variational lower bound* (ELBO):

$$\log p_\theta\big(s^{(i)}\big) \geq \mathbb{E}_{q_\phi(z|s)}\big[ -\log q_\phi(z \mid s) + \log p_\theta(s \mid z)\big] \tag{8}$$

where $\theta$ and $\phi$ are the parameters of the generation model and recognition model, respectively. Because the objective is to maximize the marginal log-likelihood, we can minimize the following loss using standard gradient-based optimization methods (Kingma & Ba, 2015):

$$\mathcal{L}_{\text{VAE}}(s; \phi, \theta) = \mathbb{E}_{q_\phi(z|s)}\big[ -\log p_\theta(s \mid z)\big] + \text{KL}\big(q_\phi(z \mid s) \,\|\, p(z)\big) \tag{9}$$

Figure 2a visualizes the probabilistic encoder-decoder model of the variational autoencoder.

## 3.2 Conditional Variational Autoencoder

As a natural extension to VAEs, Sohn et al. (2015) introduced the *conditional variational autoencoder* (CVAE), illustrating the model architecture in Figure 2b. The CVAE is comprised of a recognition (state encoder) model $q_\phi(z \mid s, \mathbf{o})$, the conditional prior (observation encoder) model $p_\theta(z \mid \mathbf{o})$, and the generation (decoder) model $p_\theta(s \mid z, \mathbf{o})$. The objective of the CVAE model is to maximize $p(s \mid \mathbf{o})$, i.e., the conditional likelihood of the data $s$ given a condition $\mathbf{o}$ (a partial observation in our case). Sohn et al. (2015) derive a conditional

extension of the variational lower bound as:

$$\log p_\theta(s \mid \mathbf{o}) \geq \mathbb{E}_{q_\phi(z|s,\mathbf{o})}\big[-\log q_\phi(z \mid s,\mathbf{o}) + \log p_\theta(s, z \mid \mathbf{o})\big] \tag{10}$$

$$= \mathbb{E}_{q_\phi(z|s,\mathbf{o})}\big[-\log q_\phi(z \mid s,\mathbf{o}) + \log p_\theta(z \mid \mathbf{o})\big] + \mathbb{E}_{q_\phi(z|s,\mathbf{o})}\big[\log p_\theta(s \mid z,\mathbf{o})\big]$$

Minimizing the following loss approximates maximizing the conditional log-likelihood:

$$\mathcal{L}_{\mathrm{CVAE}}(s, \mathbf{o}; \phi, \theta) = \mathbb{E}_{q_\phi(z|s,\mathbf{o})}\big[-\log p_\theta(s \mid z,\mathbf{o})\big] + \mathrm{KL}\big(q_\phi(z \mid s,\mathbf{o}) \,\|\, p_\theta(z \mid \mathbf{o})\big) \tag{11}$$

## 4 Inversion Variational Autoencoder

Using conditional generative models in sequential decision making problems can act as a way to sample (hidden) states conditioned on the set of observations up to time $t$. Knowing that the generative model will be used during sequential planning means we will have sets of observations from different time steps that each correspond to some true state. Therefore, we extend the CVAE model in three ways: (1) simplify the network design so that the recognition model depends only on the state, (2) directly learn the parameters $\psi$ of the conditional prior model $p_\psi(z \mid \mathbf{o})$ instead of a unit Gaussian (Sohn et al., 2015; Walker et al., 2016; Babaeizadeh et al., 2018), and (3) fine-tune the model to maximize mutual information (van den Oord et al., 2018) across observation trajectories, ensuring trajectories from the same state are closer together in the latent space and trajectories from differing states are further apart. In the EMDP setting, our belief update results in an inversion problem. Therefore, we propose the *inversion variational autoencoder* ($\mathcal{I}$-VAE) to address these challenges, illustrated in Figure 3.

The following $\mathcal{I}$-VAE loss function differs from the CVAE loss simply in the proposal $q_\phi$. The CVAE proposal conditions on both the state $s$ and observation $\mathbf{o}$, namely $q_\phi(z \mid s, \mathbf{o})$, while the $\mathcal{I}$-VAE proposal removes the dependence on the observation, namely $q_\phi(z \mid s)$. The difference lies in applying conditional independence $z \perp \mathbf{o} \mid s$ which implies that once we know the state $s$, the observation $\mathbf{o}$ provides no additional information. This simplifies the network architecture without compromising performance (see analysis in Section 6).

### 4.1 $\mathcal{I}$-VAE Pretraining Objective

The following pretraining loss function is derived in a similar manner to the CVAE loss, using the conditional variational lower bound (ELBO). To derive the $\mathcal{I}$-VAE loss function, we begin with the objective to approximate the following conditional distribution:

$$p(s \mid \mathbf{o}) = \int p(s, z \mid \mathbf{o}) \, \mathrm{d}z \tag{12}$$

Similar to the variational lower bound derivation for CVAEs (Sohn et al., 2015), we get:

$$\log p(s \mid \mathbf{o}) = \log \int p(s, z \mid \mathbf{o}) \, \mathrm{d}z \tag{13}$$

$$= \log \int p(s \mid z, \mathbf{o}) p(z \mid \mathbf{o}) \, \mathrm{d}z \tag{14}$$

$$= \log \int q_\phi(z \mid s) \frac{p(s \mid z, \mathbf{o}) p(z \mid \mathbf{o})}{q_\phi(z \mid s)} \, \mathrm{d}z \tag{15}$$

$$\approx \log \int q_\phi(z \mid s) \frac{p_\theta(s \mid z, \mathbf{o}) p_\psi(z \mid \mathbf{o})}{q_\phi(z \mid s)} \, \mathrm{d}z \tag{16}$$

$$\geq \int q_\phi(z \mid s) \log \left(\frac{p_\theta(s \mid z, \mathbf{o}) p_\psi(z \mid \mathbf{o})}{q_\phi(z \mid s)}\right) \mathrm{d}z \tag{17}$$

$$= \mathbb{E}_{q_\phi(z|s)}\left[\log\left(p_\theta(s \mid z, \mathbf{o}) \frac{p_\psi(z \mid \mathbf{o})}{q_\phi(z \mid s)}\right)\right] \tag{18}$$

$$= \mathbb{E}_{q_\phi(z|s)}\left[\log p_\theta(s \mid z, \mathbf{o})\right] + \mathbb{E}_{q_\phi(z|s)}\left[\log\left(\frac{p_\psi(z \mid \mathbf{o})}{q_\phi(z \mid s)}\right)\right] \tag{19}$$

$$= \mathbb{E}_{q_\phi(z|s)}\left[\log p_\theta(s \mid z, \mathbf{o})\right] - \mathrm{KL}\left(q_\phi(z \mid s) \,\|\, p_\psi(z \mid \mathbf{o})\right) \tag{20}$$

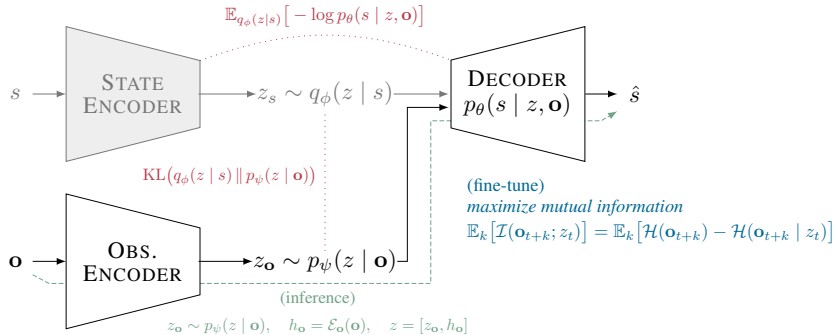

Figure 3: The inversion variational autoencoder ($\mathcal{I}$-VAE) architecture and loss.

where Equation (14) applies the chain rule, Equation (15) introduces the proposal distribution $q_\phi(z \mid s)/q_\phi(z \mid s)$, Equation (16) introduces the learnable approximations $p_\theta(s \mid z, \mathbf{o}) \approx p(s \mid z, \mathbf{o})$ and $p_\psi(z \mid \mathbf{o}) \approx p(z \mid \mathbf{o})$, Equation (17) applies Jensen's inequality, Equations (18) and (19) apply expectation and logarithmic rules, and finally Equation (20) applies the definition of KL-divergence.

To train surrogates to approximate this distribution, we want to minimize the negative log-likelihood to get the $\mathcal{I}$-VAE (pretraining) loss function:

$$\mathcal{L}_{\mathcal{I}\text{-VAE}}(s, \mathbf{o}; \phi, \theta, \psi) = \underbrace{\mathbb{E}_{q_\phi(z|s)}\big[-\log p_\theta(s \mid z, \mathbf{o})\big]}_{\text{Reconstruction loss}} + \underbrace{\text{KL}\big(q_\phi(z \mid s) \,\|\, p_\psi(z \mid \mathbf{o})\big)}_{\substack{\text{KL-divergence} \\ \text{(match latent distributions)}}} \tag{21}$$

Intuitively, the $\mathcal{I}$-VAE loss function attempts to reconstruct the state $s$ from the latent variable $z$ and match the latent conditional distributions $q_\phi(z \mid s)$ and $p_\psi(z \mid \mathbf{o})$ by minimizing their KL-divergence. Figure 4 illustrates this distribution matching. The KL-divergence term acts both as a regularizer[1] (Kingma & Welling, 2013) and as a way to learn a latent distribution that relies solely on the observations $\mathbf{o}$ during runtime inference. By penalizing their divergence, this forces both distributions to share some consistent latent structure, thus, forcing the latent distribution with partial information $\mathbf{o}$ to be close to the latent distribution with full information $s$. The CVAE model also only requires the observation during inference, but traditionally (Sohn et al., 2015; Walker et al., 2016; Babaeizadeh et al., 2018) samples $z_\mathbf{o}$ from a unit Gaussian distribution and concatenates the encoded observation $z_\mathbf{o}$ to get $z = [z_\mathbf{o}, h_\mathbf{o}]$, where $h_\mathbf{o} = \mathcal{E}_\mathbf{o}(\mathbf{o})$ is the encoded observation and $z$ is passed to the decoder. The $\mathcal{I}$-VAE model explicitly learns an approximate conditional model $p_\psi(z \mid \mathbf{o})$, which uses the observation directly to shape where in the latent space to sample from. This results in matching the inference-time behavior with the training-time behavior.

Our work is related to the *variational autoencoder with arbitrary conditioning* (VAEAC) (Ivanov et al., 2019), but their model assumes the observed data is simply a mask of the hidden (full) data. This is true for pixel observations from images (e.g., MNIST (LeCun et al., 1998), shown in Figure 5), but breaks down in cases where the observations are over a different domain (e.g., muon tomography observations). Our model does not make this assumption about the observations and we show that it can work in both cases.

**Latent space mapping and runtime inference.** Illustrated in Figure 5, the goal during pretraining is to map both the latent-state variable $z_s$ and latent-observation variable $z_\mathbf{o}$ to a lower-dimensional representation in $\mathbb{R}^d$. Using the KL-divergence from Equation (21), we expect these latent variables to map to a similar area in the lower $d$-dimensional space. During runtime inference, the reduced-dimension latent $z_\mathbf{o}$ is used to sample states from $p_\theta(s \mid z, \mathbf{o})$ consistent with the observations. Figure 5 shows an example using masked $4 \times 4$ pixel observations for the MNIST dataset, where at time $t = 30$ the total pixel coverage given by the observations is about 15%. A set of state samples $\mathbf{s}$ can be directly used to represent samples from an updated belief in EMDP planning.

---

[1]The KL-divergence stops $q_\phi(z \mid s)$ from mode-collapsing or becoming arbitrarily tight around each training sample $s$, and it prevents $p_\psi(z \mid \mathbf{o})$ from ignoring the observation $\mathbf{o}$ or becoming too broad.

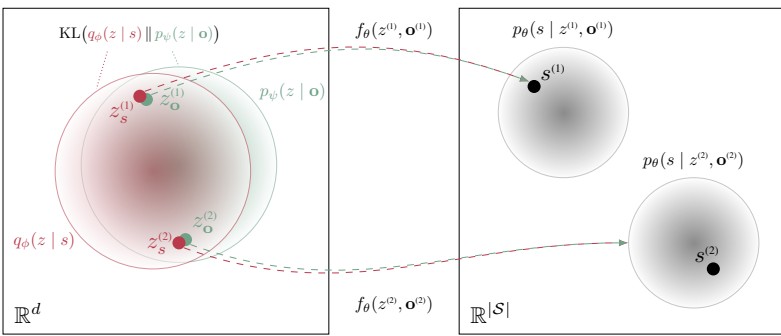

Figure 4: Latent space matching of $z_s$ and $z_\mathbf{o}$ and mapping to $p_\theta(s \mid z, \mathbf{o})$ for the $\mathcal{I}$-VAE.

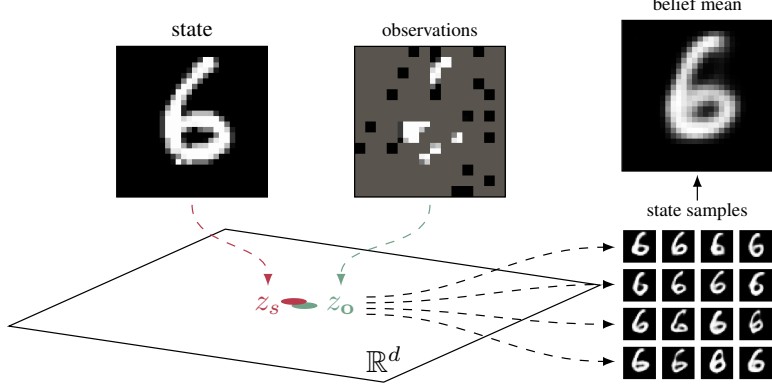

Figure 5: Latent space mapping of $z_\mathbf{o}$ ($t = 30$) to $k = 100$ state samples from $p_\theta(s \mid z, \mathbf{o})$.

## 4.2 Fine-Tuning to Maximize Trajectory-Based Mutual Information

Not only are we interested in modeling $p(s \mid \mathbf{o})$ from Equation (12), we are also in the setting where sequences of observations will be passed to the model over time. Thus, a secondary objective is to maximize mutual information between observation sequences (i.e., observation trajectories), to better align the latent space across observations correlated to the same state. Therefore, we fine-tune the pretrained $\mathcal{I}$-VAE model with a *trajectory contrastive loss* (TCL) (van den Oord et al., 2018; Halawa et al., 2022; Zheng et al., 2023). The key idea is that once we have a performant generative model with trained encoders and decoder, we can further align the latent space given observation trajectories from a dataset $(s, \mathbf{o}_{1:T}) \in \mathcal{D}_{\text{traj}}$ of states and the full set of observations over the time horizon $T$. Namely, $\mathbf{o}_{1:T} = (\mathbf{o}_1, \mathbf{o}_2, \ldots, \mathbf{o}_T)$, where $\mathbf{o}_t$ is equivalent to $o_{1:t}$, i.e., the incomplete observation at time $t$.

The trajectory contrastive loss is a form of the InfoNCE loss (van den Oord et al., 2018), where we want to maximize mutual information $\mathcal{I}$ between a window of $k \in [1, \ldots, K]$ observations and the observation-inferred latent variables $z_t$. Namely, we want to maximize the expectation of $\mathcal{I}(\mathbf{o}_{t+k}; z_t)$ over the $K$-length window, or minimize the negation:

$$\mathcal{L}_{\text{TCL}}(\mathbf{o}_{1:T}; \omega, K) = -\mathbb{E}_k[\mathcal{I}(\mathbf{o}_{t+k}; z_t)] \tag{22}$$

$$= -\mathbb{E}_k[\mathcal{H}(\mathbf{o}_{t+k}) - \mathcal{H}(\mathbf{o}_{t+k} \mid z_t)] \quad \text{for} \quad 1 \leq k \leq K \tag{23}$$

$$= -\mathbb{E}_k\left[\mathbb{E}_{z_t}\left[\log \frac{p(\mathbf{o}_{t+k} \mid z_t)}{p(\mathbf{o}_{t+k})}\right]\right] \tag{24}$$

$$\geq -\mathbb{E}_k\left[\mathbb{E}_{\tilde{z}}\left[\log \frac{f(\mathbf{o}^+, \tilde{z})}{\sum_{\mathbf{o} \in \Omega^-} f(\mathbf{o}, \tilde{z})}\right]\right] \tag{25}$$

where $\mathcal{I}(a, b)$ represents the mutual information between $a$ and $b$, entropy is denoted by $\mathcal{H}$, the space $\Omega^- = \{\Omega \setminus \mathbf{o}^+\}$ is over all observation sequences excluding the positive observation $\mathbf{o}^+$, the latent $\tilde{z} = \Pi_\omega(\mu_{\mathbf{o}_t})$ is the output of a projection network parameterized by $\omega$, and the function $f$ measures the similarity between the observation $\mathbf{o}_t$ and projected latent variable $\tilde{z}$, following van den Oord et al. (2018):

$$f(\mathbf{o}_t, \tilde{z}) = \exp\left(\frac{\tilde{z}^\top h_t}{\|\tilde{z}\|\|h_t\|}/\tau\right) \quad \text{where} \quad h_t = \mathcal{E}_\mathbf{o}(\mathbf{o}_t) \tag{26}$$

where $\tau$ is a temperature parameter. For a window of length $K$, by minimizing the loss $\mathcal{L}_{\text{TCL}}$, we maximize mutual information for observation trajectories from $\mathbf{o}_t$ to $\mathbf{o}_{t+K}$. The idea is that we want to optimize the latent space so that observations that come from the same sequence are closer together, while observations from different sequences are further apart. In addition to fine-tuning the mutual information, we also want to preserve reconstruction and generation. Therefore, the fine-tuning $\mathcal{I}$-VAE loss function then becomes:

$$\mathcal{L}(s, \mathbf{o}_{1:T}; \omega, \theta, \phi, \psi, K) = \alpha\mathcal{L}_{\text{TCL}}(\mathbf{o}_{1:T}; \omega, K) + \mathbb{E}_t\left[\mathcal{L}^{(t)}_{\mathcal{I}\text{-VAE}}(s, \mathbf{o}_t; \theta, \phi, \psi)\right] \tag{27}$$

The TCL loss acts as a self-supervised consistency loss on the observation encoder, helping it produce a smooth, predictive latent path (Zheng et al., 2023). In practice, we train the projection network $\Pi_\omega$ to discriminate between observation trajectories.

**Training and inference.** The $\mathcal{I}$-VAE training procedure and forward pass are detailed in Algorithms 1 and 2. During runtime inference, the input observation is encoded to get $h_\mathbf{o} = \mathcal{E}_\mathbf{o}(\mathbf{o})$, a latent-observation variable $z_\mathbf{o}$ is sampled from $p_\psi(z \mid \mathbf{o})$, and a sampled state is reconstructed from the decoder $p_\theta(s \mid z, \mathbf{o})$ using $z = [z_\mathbf{o}, h_\mathbf{o}]$.

---

**Algorithm 1** $\mathcal{I}$-VAE training procedure.

---

**Require:** $\theta, \phi, \psi, \omega \leftarrow$ initialize network parameters
**Require:** $\alpha \leftarrow$ trajectory contrastive loss weighting
**Require:** $\tau \leftarrow$ trajectory contrastive loss temperature
**Require:** $T \leftarrow$ observation horizon

1:   **for** epoch $\leftarrow 1$ **to** $n_{\text{pretrain}}$
2:      Sample $(s, \mathbf{o})$ from $\mathcal{D}_{\text{partial}}$
3:      Forward pass $(s, \mathbf{o})$ to get $(\hat{s}, \mu_s, \log\sigma_s^2, \mu_\mathbf{o}, \log\sigma_\mathbf{o}^2)$ calling algorithm 2
4:      Compute loss $\mathcal{L}_{\mathcal{I}\text{-VAE}}(s, \mathbf{o}; \theta, \phi, \psi)$ given $(\hat{s}, \log\sigma_s^2, \mu_\mathbf{o}, \log\sigma_\mathbf{o}^2)$ with eq. (21)
5:      Train $\theta$, $\phi$, and $\psi$

6:   **for** epoch $\leftarrow 1$ **to** $n_{\text{fine-tune}}$
7:      Sample $(s, \mathbf{o}_{1:T})$ from $\mathcal{D}_{\text{traj}}$
8:      Compute $\mu_s$ and $\log\sigma_s^2$ from state encoder $\mathcal{E}_s(s)$
9:      **for** $t \leftarrow 1$ **to** $T$
10:        Encode $h_{\mathbf{o}_t}$ from observation encoder $\mathcal{E}_\mathbf{o}(\mathbf{o}_t)$ using $\mathbf{o}_t \in \mathbf{o}_{1:T}$
11:        Compute $\mu_{\mathbf{o}_t}$ and $\log\sigma_{\mathbf{o}_t}^2$ from $p_\psi(z \mid \mathbf{o}_t)$ using observation encoding $h_{\mathbf{o}_t}$
12:        Project $\tilde{z} = \Pi_\omega(\mu_{\mathbf{o}_t})$
13:        Decode $\hat{s}_t$ from $p_\theta(s \mid \tilde{z}, \mathbf{o})$
14:        Compute $\mathcal{L}_t = \mathcal{L}_{\mathcal{I}\text{-VAE}}(s, \mathbf{o}_t; \theta, \phi, \psi)$ from eq. (21) using $\hat{s}_t$
15:      Compute combined loss $\mathcal{L} = \alpha\mathcal{L}_{\text{TCL}}(\mathbf{o}_{1:T}; \omega, K) + \mathbb{E}_t[\mathcal{L}_t]$ using eq. (22)
16:      **if** epoch $= 1$
17:        Train projection network parameters $\omega$
18:      **else**
19:        Train $\omega$ and $\psi$ (optional: $\theta$ and $\phi$)

---

---

**Algorithm 2** $\mathcal{I}$-VAE forward pass.

---

**Require:** $s, \mathbf{o}$ (state and observations)
  1. Compute $h_s$, $\mu_s$, and $\log \sigma_s^2$ from state encoder $h_s = \mathcal{E}_s(s)$ and $q_\phi(z \mid s)$
  2. Sample $z \sim q_\phi(z \mid s)$ using state encoding $h_s$
  3. Compute $h_\mathbf{o}$, $\mu_\mathbf{o}$, and $\log \sigma_\mathbf{o}^2$ from observation encoder $h_\mathbf{o} = \mathcal{E}_\mathbf{o}(\mathbf{o})$ and $p_\psi(z \mid \mathbf{o})$
  4. Decode $\hat{s}$ from $p_\theta(s \mid z, \mathbf{o})$ using $z$ and observation encoding $h_\mathbf{o}$
  5. **return** $(\hat{s}, \mu_s, \log \sigma_s^2, \mu_\mathbf{o}, \log \sigma_\mathbf{o}^2)$

---

## 5 Batched Belief-State Markov Decision Process

To efficiently use generative inversion methods for planning, which are naturally parallelized by design, we must define a decision making framework that supports parallelism. Given a partially observable Markov decision process (POMDP) defined by the tuple $\langle \mathcal{S}, \mathcal{A}, \mathcal{O}, T, R, O, \gamma \rangle$, we can convert the POMDP to an equivalent belief-state MDP (see Figure 6). Using the batched planning model derived in Appendix Section A.2, we can define the *batched belief-state MDP* (**BB**-MDP) framework by treating a batch of beliefs $\mathbf{b} = (b_i)_{i=1}^m$ as states (shown in Figure 7). The **BB**-MDP is specified by the tuple $\langle \mathcal{B}^m, \mathcal{A}^m, \mathbf{T}_b, \widehat{R}_b, \gamma \rangle$, where the batched belief-state transition function $\mathbf{b}' \sim \mathbf{T}_b(\cdot \mid \mathbf{b}, \mathbf{a})$ is defined as $\mathbf{T}_b : \mathcal{B}^m \times \mathcal{A}^m \to \Delta(\mathcal{B}^m)$, internally computing the following four steps:

$$\mathbf{s} = (s_i)_{i=1}^m \quad \text{where} \quad s_i \sim b_i \quad \text{for each} \quad b_i \in \mathbf{b} \tag{28}$$

$$\mathbf{s}' = (s_i')_{i=1}^m \quad \text{where} \quad s_i' \sim T(\cdot \mid s_i, a_i) \quad \text{for each} \quad (s_i, a_i) \in (\mathbf{s}, \mathbf{a}) \tag{29}$$

$$\mathbf{o} = (o_i)_{i=1}^m \quad \text{where} \quad o_i \sim O(\cdot \mid a_i, s_i') \quad \text{for each} \quad (s_i', a_i) \in (\mathbf{s}', \mathbf{a}) \tag{30}$$

$$\mathbf{b}' = (b_i')_{i=1}^m \quad \text{where} \quad b_i' \leftarrow \textsc{Update}(b_i, a_i, o_i) \quad \text{for each} \quad i \in \{1, \ldots, m\} \tag{31}$$

where $b \in \mathcal{B}$ is a belief state belonging to the belief simplex $\mathcal{B}$ over the state space $\mathcal{S}$, and the batched belief-state space $\mathcal{B}^m$ is the space of ordered $m$-tuples of beliefs:

$$\mathcal{B}^m = \Big( (b_1, \ldots, b_m) \mid b_i \in \mathcal{B} \Big) \tag{32}$$

with $\mathbf{b} \in \mathcal{B}^m$ denoting a batched belief-state. If the four steps of the batched belief transition can be done in parallel over batches (i.e., the state sampling procedure from the belief, the state-transition function,

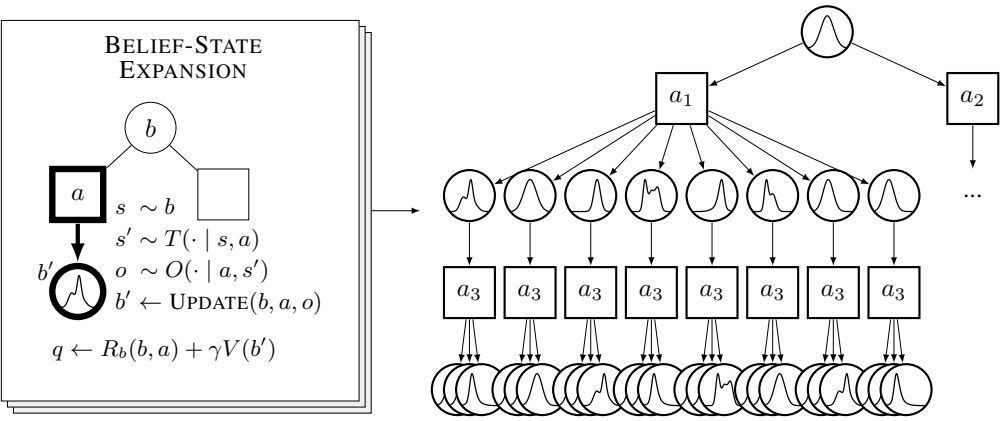

Figure 6: Belief-state MDP for two-step expansion.

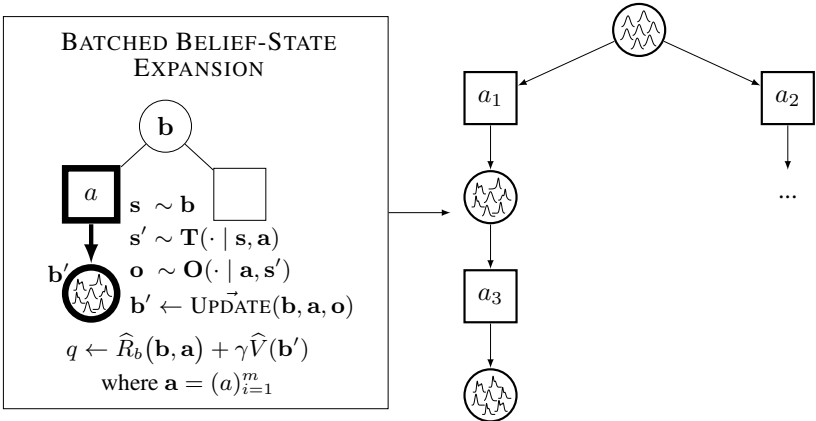

Figure 7: Batched belief-state MDP for two-step expansion.

observation function, and belief update), then Equations (28) to (31) can be simplified using the following notation:

$$\mathbf{s} \sim \mathbf{b} \qquad \mathbf{s}' \sim \mathbf{T}(\cdot \mid \mathbf{s}, \mathbf{a}) \qquad \mathbf{o} \sim \mathbf{O}(\cdot \mid \mathbf{a}, \mathbf{s}') \qquad \mathbf{b}' = \overrightarrow{\text{UPDATE}}(\mathbf{b}, \mathbf{a}, \mathbf{o}) \qquad (33)$$

for the vectorized transition function $\mathbf{T} : \mathcal{S}^m \times \mathcal{A}^m \to \Delta(\mathcal{S}^m)$, the vectorized observation function $\mathbf{O} : \mathcal{A}^m \times \mathcal{S}^m \to \Delta(\mathcal{O}^m)$, and the vectorized belief update function $\overrightarrow{\text{UPDATE}} : \mathcal{B}^m \times \mathcal{A}^m \times \mathcal{O}^m \to \mathcal{B}^m$, where the particle instantiation is a stochastic approximation of UPDATE. The batched belief reward function is then derived from Equation (56):

$$\widehat{R}_b(\mathbf{b}, \mathbf{a}) = \mathop{\mathbb{E}}_{b \in \mathbf{b}, \, a \in \mathbf{a}} \left[ R_b(b_i, a_i) \right] = \frac{1}{m} \sum_{i=1}^{m} \left( \int b_i(s) R(s, a_i) \, \mathrm{d}s \right) \qquad (34)$$

where $R_b$ is the belief-state reward function.

## 5.1 Batched Belief-State Value Function

To act optimally under partial observability, an agent must evaluate the expected return of a policy $\pi$ given its current belief $b \in \mathcal{B}$. The belief-state value function is then the expectation of state values weighted by the likelihood under the belief (Kaelbling et al., 1998):

$$V^\pi(b) = \int b(s) V^\pi(s) \, \mathrm{d}s \qquad (35)$$

Extending to the batched belief-state model and using the decomposition in Theorem 1 outlined in Appendix Section A.4, we define the batched belief-state value function as the average value across a batch of beliefs:

$$\widehat{V}^{\boldsymbol{\pi}}(\mathbf{b}) = \frac{1}{m} \sum_{i=1}^{m} \left( \int b_i(s) V^\pi(s) \, \mathrm{d}s \right) \qquad (36)$$

This value reflects the expected return from executing the policy $\pi$ independently from each belief in the batch, and serves as the basis for value-based planning under state uncertainty. We can similarly show that when $\pi$ is the optimal policy $\pi^*$, then $\widehat{V}^\pi = \widehat{V}^{\pi^*}$ (following the same argument as Theorem 1).

In the underlying POMDP, the optimal belief-state $Q$-value function is similarly defined as the expected $Q$-value under the belief:

$$Q^*(b, a) = \int b(s) Q^*(s, a) \, \mathrm{d}s \qquad (37)$$

Extending to the batched planning model, we define the batched belief-state $Q$-value function as the average over the batch of beliefs:

$$\widehat{Q}^*(\mathbf{b}, a) = \frac{1}{m} \sum_{i=1}^{m} Q^*(b_i, a) = \frac{1}{m} \sum_{i=1}^{m} \left( \int b_i(s) Q^*(s, a) \, ds \right) \tag{38}$$

This function evaluates the expected return of executing action $a$ across all belief points in the batch $\mathbf{b}$. The corresponding optimal batched belief-state policy vector is comprised of the actions that maximizes the optimal $Q$-value:

$$\boldsymbol{\pi}^*(\mathbf{b}) = \left( \arg\max_{a \in \mathcal{A}} Q^*(b_i, a) \mid b_i \in \mathbf{b} \right) \tag{39}$$

**Action selection from a single belief.** In the outer POMDP loop when we have a single belief $b$, following Appendix Section A.3, we set $\mathbf{b} = (b)_{i=1}^{m}$ as copies of the current belief $b$ to allow the batched planning model to expand in parallel from the current belief. The action selected at the outer POMDP loop then becomes:

$$\pi^*(b) = \arg\max_{a \in \mathcal{A}} \widehat{Q}^*(\mathbf{b}, a) \quad \text{for} \quad \mathbf{b} = (b)_{i=1}^{m} \tag{40}$$

This policy selects a single action that performs well in expectation over the batch of belief states (and their future expansion), enabling robust decision making under state uncertainty.

## 6 Experiments and Analysis

Figure 8 shows how sequential scaling compares to batched scaling on an 80GB NVIDIA A100 GPU. The figure shows an aggregate batch for $n = |\mathcal{A}|$ actions, $m = |\mathbf{b}|$ beliefs per action, and $k = |b|$ states per belief, where we represent each belief $b$ as a collection of state particles $b = (s_i)_{i=1}^{k}$. Unsurprisingly, the results indicate that sequential scaling grows rapidly with batch size, while batched GPU scaling remains efficient.

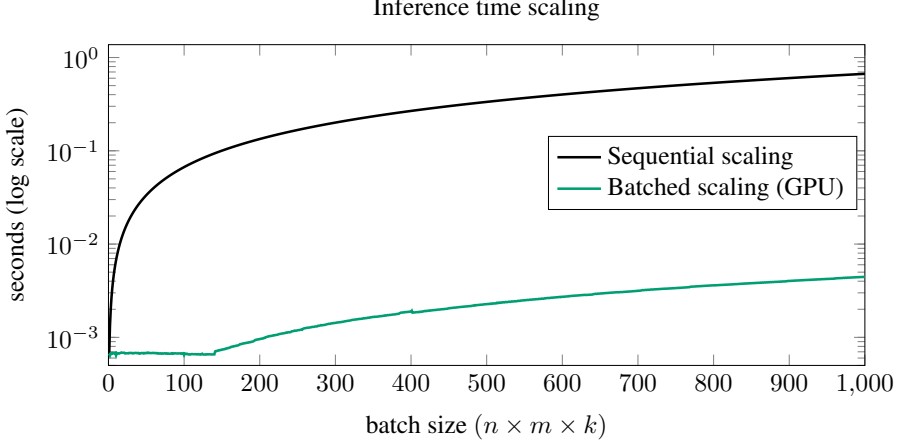

Figure 8: Inference time scaling in **BB**-MDPs.

We demonstrate the effectiveness of the $\mathcal{I}$-VAE on two sequential problems: partial MNIST observations (LeCun et al., 1998) and a real-world muon tomography mineral exploration EMDP. The model code,[2] written in PyTorch, and the muon observation EMDP code,[3] written in Julia and Python, are open-sourced to extend to other problems and for reproducibility. We further demonstrate performance against a tractable posterior, and baseline against a conditional diffusion model, in a Gaussian-mixture inverse problem in Appendix Section A.8. Details regarding hyperparameters and neural network architectures are included in the open-sourced code.

---

[2]https://github.com/anonymous-algodev/I-VAE
[3]https://github.com/anonymous-algodev/MuonPOMDPs.jl

## 6.1   Pixel Observation Problem: MNIST

Using the standard MNIST benchmark of hand-written digits, we are interested in measuring how well the model performs given partial observations. We partition the image into 196 chunks of $4 \times 4$ pixels each to act as individual observations. The MNIST state is represented as a finite-dimensional grayscale image $s \in [0,1]^D$, where coordinate $d$ indexes one of the $D$ pixels. In this problem, the observations can be interpreted as a mask applied to the true (hidden) state. Given random pixel observations over time, we refer to the "unmasked" observations as the observation coverage percentage (e.g., if 49 out of 196 chunks are uncovered, we have 25% coverage). We compare the $\mathcal{I}$-VAE model against a standard CVAE, a deterministic neural network (NN) baseline, and against the $\mathcal{I}$-VAE pretrained model (i.e., without fine-tuning with TCL). All of the models are simple MLPs following parameters used by Sohn et al. (2015) for the original MNIST CVAE experiments. To measure the estimation quality of the models, we evaluate the conditional log-likelihood (CLL) across each observation coverage percentage:

$$\log p_\theta(s \mid \mathbf{o}) \approx \log \left( \frac{1}{n} \sum_{i=1}^n p_\theta\big(s \mid z^{(i)}, \mathbf{o}\big) \right) \quad \text{where} \quad z^{(i)} \sim p_\psi(z \mid \mathbf{o}) \tag{41}$$

where $n$ is the size of the MNIST test dataset ($n = 10{,}000$). The CLL metric is a standard way to evaluate the statistical performance of the models.

Figure 9 shows the conditional log-likelihoods across the various models for different observation coverage percentages. Unsurprisingly, the neural network baseline performs the worst as it will deterministically infer a single state given the observations. In the MNIST case, we observe that the $\mathcal{I}$-VAE without TCL performs just as well as the CVAE model. The benefit of additional TCL fine-tuning is most evident in regimes with low observation coverage (i.e., around 30% observation coverage or less). In this regime, the $\mathcal{I}$-VAE model achieves the highest conditional log-likelihood, indicating its ability to infer missing pixel values from sparse observations. This suggests that the learned latent prior in the $\mathcal{I}$-VAE captures a more informative and observation-aligned structure of the data, enabling more accurate posterior sampling when only partial information is available. We find that TCL improvement is robust to choice of TCL weighting parameter $\alpha$, and present an ablation in Appendix Section A.9.

Notably, the $\mathcal{I}$-VAE model without fine-tuning with the TCL objective (dotted lines) closely tracks the CVAE performance, reinforcing that the primary improvement stems from the inversion architecture itself, while TCL provides an additional gain in low-coverage areas. Beyond this range, as observation coverage increases and the data becomes more complete, the difference between models narrows, and all approaches converge toward similar log-likelihoods as the inference task becomes easier.

Shown next, both $\mathcal{I}$-VAE models outperform the CVAE when using an information-based heuristic planner, framing the sequential information gathering problem as an EMDP.

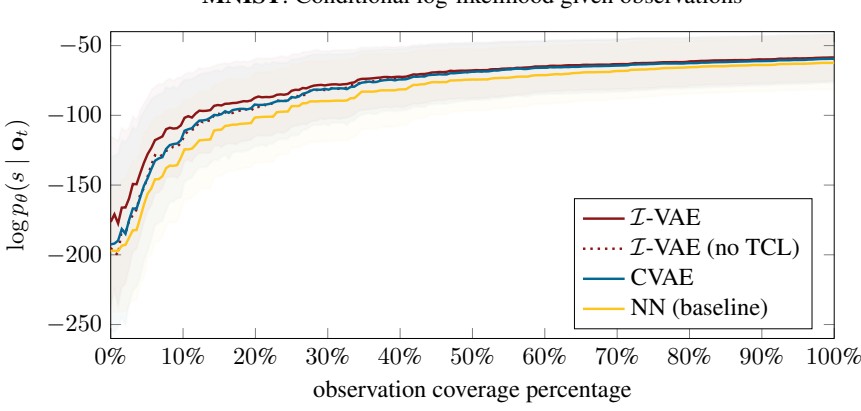

Figure 9: Conditional log-likelihood over the MNIST test dataset ($\pm$ one stddev).

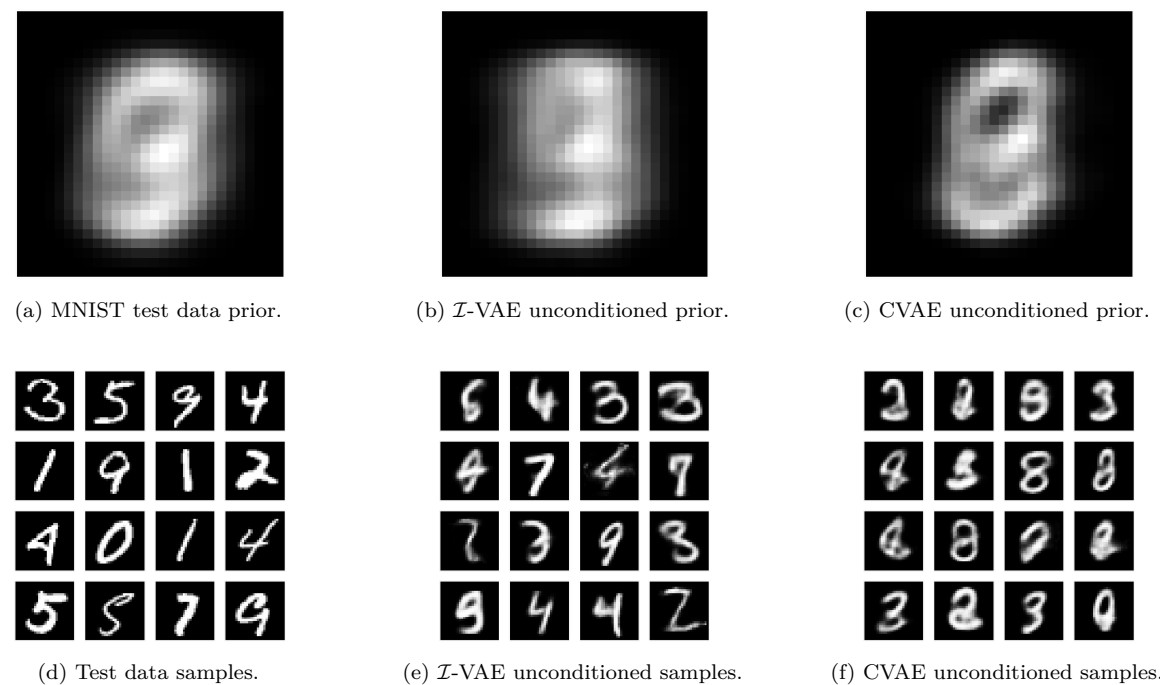

(a) MNIST test data prior.   (b) $\mathcal{I}$-VAE unconditioned prior.   (c) CVAE unconditioned prior.

(d) Test data samples.   (e) $\mathcal{I}$-VAE unconditioned samples.   (f) CVAE unconditioned samples.

Figure 10: Mean priors (i.e., no observations), sampling $k = 1000$ for $\mathcal{I}$-VAE and CVAE.

Figure 10 qualitatively illustrates how the $\mathcal{I}$-VAE can better reconstruct the data when conditioning on no observations (i.e., observation **o** is solely comprised of all $-1$ values). In this unconditioned case, when the observation coverage is 0%, we can visually see that when sampling states with no observation information, the mean belief, shown in Figures 10a to 10c, matches closer to the MNIST test data prior. We also show reconstructed samples, conditioned without any observation information in Figures 10d to 10f. We provide additional qualitative analysis and Figures 19 and 20 demonstrating that $\mathcal{I}$-VAE converges to the correct state in fewer observations than CVAE, in the appendix.

## 6.2  Indirect Observation Problem: Muon Tomography

In the previous section, we showed how the $\mathcal{I}$-VAE model operates when treating pixel observations as a state mask. In more realistic geological problems, different measurement sources, such as seismic imaging (Deng et al., 2022) or muon tomography (Schultz et al., 2007; Lechmann et al., 2021; Schouten et al., 2022) (shown in Figure 11), are used to infer the unknown state of the subsurface. Statistical models of the subsurface are particularly useful as they capture the uncertainty in what's underground (Kaipio & Somersalo, 2006). The muon ground-truth state is represented as a finite-dimensional binary occupancy field $s \in \{0, 1\}^D$, where coordinate $d$ indexes one voxel of the subsurface intrusion field. The learned generative belief updater outputs relaxed occupancy values $\hat{s} \in [0, 1]^D$, and belief means and variances are computed over these generated state particles.

In the field of mineral exploration (Haldar, 2018), typical strategies consist of drilling or placing sensors in an exhaustive grid to collect subsurface information. More recently, Mern et al. (2021) framed the mineral exploration problem as a POMDP and used state-of-the-art POMDP solvers to intelligently select where to drill, avoiding standard grid-based policies. Mern et al. (2021) treated drill actions as observing borehole data to collect subsurface information. Mineral exploration is a multi-objective problem: we want to maximize information gain while simultaneously minimizing the overall cost (e.g., minimizing the number of drills). The ultimate objective of the mineral exploration problem is to make a `go/no-go` decision to `mine` or `abandon` based on some economic value of the subsurface intrusion (e.g., the mass of a subsurface ore body being worth the cost to extract). The problem can be framed as an information gathering EMDP where the true state does not transition, therefore, the state is static.

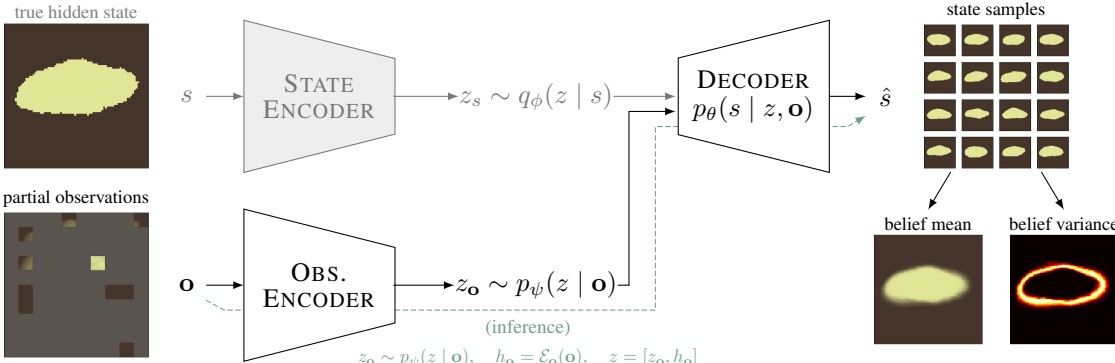

Figure 11: Muon observation inversion process using an $\mathcal{I}$-VAE with 10 observations.

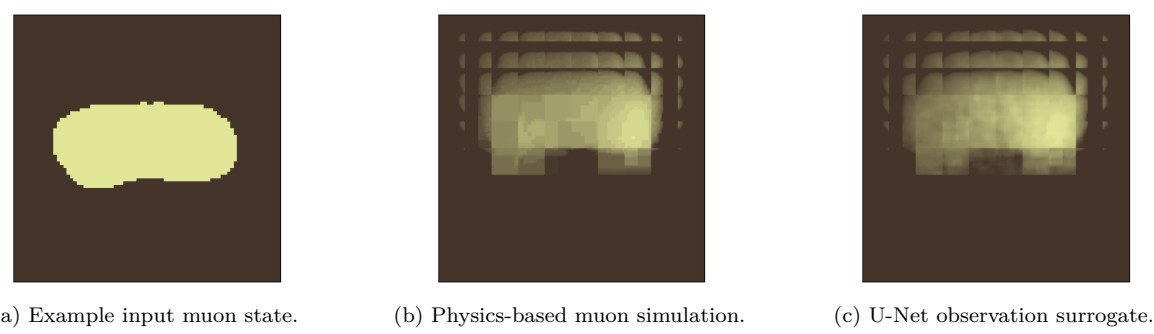

(a) Example input muon state.  (b) Physics-based muon simulation.  (c) U-Net observation surrogate.

Figure 12: Muon density U-Net surrogate model compared to the physics-based simulator.

### 6.2.1 Sequential Information Gathering using Batched Belief-State Planning

Crucial to the success of the POMDP solvers is the efficacy of the belief updater. Mern et al. (2021) used a particle filter given a known observation model $O(o \mid a, s')$. In our work, we study the case where the observation model is not known (or difficult to compute) and where observations are muon tomography measurements from sensors placed in boreholes, and drill actions place a set of 9 passive muon sensors into the subsurface, as shown in 3D in Figure 1. The muon tomography EMDP receives a penalty for every drill action executed, a reward proportional to the difference in the extracted intrusion mass compared to an economic threshold if the `go` action is taken, and zero reward in the case of a `no-go` decision.

Introduced in Section 5, we convert the mineral exploration with muon tomography EMDP (termed *muon-based intrusion discovery*) into its batched equivalent. Aside from the observation function (which we cover

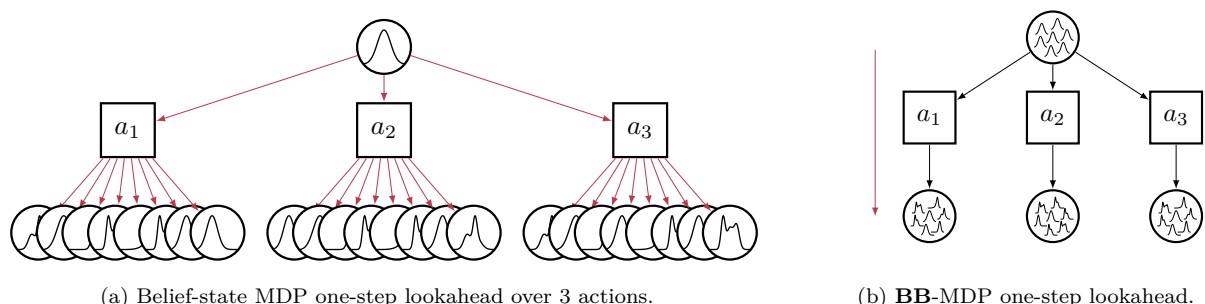

(a) Belief-state MDP one-step lookahead over 3 actions.  (b) **BB**-MDP one-step lookahead.

Figure 13: Indicating number of belief transition calls in red for BDMPs vs. **BB**-MDPs.

next), the $\mathcal{I}$-VAE belief updater is designed to be parallel and, trivially, there are no state-transition dynamics. As for the forward observation function, we trained a simple U-Net (Ronneberger et al., 2015) as a surrogate observation model (Figure 12) using data from a physics-based muon tomography simulator. Therefore, we can directly convert the muon-based intrusion discovery problem into a **BB**-MDP. Using this formulation, along with the neural network surrogates, we can sample $k$ state particles and evaluate all 100 actions (corresponding to muon sensor drill locations) in one forward pass to the U-Net observation surrogate and one forward pass to the $\mathcal{I}$-VAE belief updater as a batch, shown compared to the standard belief-state MDP (BMDP) in Figure 13.

**Information-based heuristic policy.** To test the performance of the $\mathcal{I}$-VAE belief updater in the batched planning setting, and as a tractable approximation to Problem 1, we use an information-based heuristic policy to maximize expected Bayesian information gain. At the distribution level, the ideal one-step objective is the expected entropy reduction:

$$\mathcal{I}(a;b) = \mathcal{H}(b) - \mathbb{E}_i\left[\mathcal{H}(b_i')\right] \quad \text{where} \quad b' = (b_1', \ldots, b_m') \sim T_b(\cdot \mid b, a). \tag{42}$$

Here $\mathbb{E}_i[\cdot]$ denotes the expectation over possible updated beliefs induced by action $a$, approximated in the **BB**-MDP by the empirical average over $m$ sampled updated beliefs. Theorem 2 shows that maximizing this distribution-level expected entropy reduction is equivalent to maximizing expected KL-divergence, which is related to the *efficacy of information* (EOI) (Caers et al., 2022).

In practice, for image- and voxel-valued particle beliefs, we estimate entropy using a marginal particle entropy estimator. For a particle belief $b = (s_1, \ldots, s_k)$ with $s_j \in [0,1]^D$, we define:

$$\widehat{\mathcal{H}}_{\mathrm{marg}}(b) = \frac{1}{D}\sum_{d=1}^{D} h\left(\frac{1}{k}\sum_{j=1}^{k} s_j[d]\right) \tag{43}$$

where $h$ is the binary entropy function. The empirical information score used by the heuristic is then:

$$\widehat{\mathcal{I}}_{\mathrm{marg}}(a;b) = \widehat{\mathcal{H}}_{\mathrm{marg}}(b) - \frac{1}{m}\sum_{i=1}^{m} \widehat{\mathcal{H}}_{\mathrm{marg}}(b_i') \tag{44}$$

Here $b'$ is an $m$-sized batch of candidate updated beliefs, each represented with $k$ state particles. We use $m = 3$ beliefs in the batch and $k = 100$ state samples per belief.

The information-based heuristic policy will therefore select actions that maximize the information gain, then make a final decision based on the probability that the economic intrusion volume is above some risk threshold $1 - \Delta$. The heuristic policy is defined as:

$$\pi_{\mathrm{heuristic}}(b) = \begin{cases} \texttt{go} & \text{if } P(v > 0) > 1 - \Delta \\ \texttt{no-go} & \text{if } P(v < 0) > 1 - \Delta \\ \arg\max_{a \in \mathcal{A}} \widehat{\mathcal{I}}_{\mathrm{marg}}(a;b) & \text{otherwise} \end{cases} \tag{45}$$

where the distribution over intrusion volume $v$ is computed over the belief $b$ and standardized so that the nominal (prior) volume has zero mean. When the standardized volume is above zero, this indicates the intrusion is economical to mine. This way, if we randomly took `go`/`no-go` actions, we would make the correct decision 50% of the time. The parameter $\Delta$ controls the risk tolerance in the final `go`/`no-go` decision. In the following experiments, we set $\Delta = 0.1$, resulting in making the final decision if the confidence in our estimated volume distribution is above 90%. Other criteria, such as the upper-confidence bound (UCB1) algorithm (Auer et al., 2002) or Bayes-UCB (Kaufmann et al., 2012), could be used instead.

**Baseline policies.** Along with the heuristic policy, we evaluate four other baseline policies. The first two baseline policies select actions deterministically following sweeping grid-like patterns. Starting from the top-left origin of the $10 \times 10$ action space, the *horizontal grid* policy selects actions in a right-to-left, then

left-to-right sweeping pattern. Similarly, the *vertical grid* policy selects actions in a downward, then upward pattern; mimicking conventional drilling strategies (Haldar, 2018). We also baseline against a *random* policy that will randomly select actions uniformly over the action space. Finally, given privileged information about the true state under test, we test against an *oracle* that selects the action that minimizes the error between the mean posterior belief and the true state (with access to the ground truth state):

$$\pi_{\text{oracle}}(b) = \begin{cases} \texttt{go} & \text{if } P(v > 0) > 1 - \Delta \\ \texttt{no-go} & \text{if } P(v < 0) > 1 - \Delta \\ \underset{a \in \mathcal{A}}{\arg\min} \left| \mathbb{E}[b'] - s_{\text{true}} \right| & \text{otherwise} \end{cases} \tag{46}$$

The oracle policy determines the best performance that each belief updater could achieve. We compare the $\mathcal{I}$-VAE with and without TCL fine-tuning, the CVAE model, and a particle filter with reinvigoration where the estimated observation likelihood measures how close each observation is to a sampled state from the belief, proportional to the mean-squared error between the observation generated from the sampled state and the environment observation.

**Planning and estimation metrics.** We use several metrics to compare how well the $\mathcal{I}$-VAE estimates the conditional likelihood $p(s \mid \mathbf{o})$ and its performance during planning. For estimation quality, we use the conditional log-likelihood defined in Equation (41). To evaluate the planning performance using the heuristic and baseline policies, we measure the accuracy in the final `go`/`no-go` decision (where `go` should be executed if the economic volume of a test intrusion is above zero), and we measure how many actions it took to reach the final decision. Finally, we measure the error in the belief compared to the true test state over different number of actions/observations.

### 6.2.2 Muon-Based Intrusion Discovery Results and Analysis

Table 1 and Figure 14 show the estimation performance of the neural network belief surrogates. When selecting drill actions, each surrogate uses the information-based heuristic policy *without* making a final decision (to ultimately evaluate the performance over all drill actions). Evident in our experiments, we see that $\mathcal{I}$-VAE achieves better estimation when compared to CVAE and $\mathcal{I}$-VAE without TCL. The confidence band overlap in Figure 14 is more pronounced at higher observation coverage. The TCL benefit concentrates in the low-coverage regime (below 30% in MNIST), where separation between the methods is visible in the figures. This is precisely the regime most relevant to planning, where early decisions must be made with sparse observations. Unlike the MNIST example, in the muon tomography case, the initial belief (i.e., conditioned on no observations) is more diffuse, resulting in worse early estimation performance.

Regarding planning performance, Table 2 indicates that the $\mathcal{I}$-VAE model produces the highest accuracy in the fewest number of actions. We also highlight that the additional TCL fine-tuning step improves the base model. Accuracy and action count results for $\mathcal{I}$-VAE with and without TCL are independently presented in bold when statistically significant relative to both particle filter and CVAE methods. Comparing the belief updating results in Table 2 to their respective oracle column shows the theoretically one-step optimal performance achievable by each method. This is also evident when looking at the belief error in Figure 15 and comparing the $\mathcal{I}$-VAE heuristic to the $\mathcal{I}$-VAE oracle, showing that the information-based heuristic is approximately optimal. It is also clear that the particle filter performs poorly on this problem as it suffers from particle depletion (i.e., the belief collapses). This is due to the fact that the particle filter initially

Table 1: Muon problem: CLLs given different number of observations.

| Method | Conditional log-likelihood (CLL) | | | | |
|---|---|---|---|---|---|
| | 0 | 25 | 50 | 75 | 100 |
| CVAE | $-563.41$ | $-213.98$ | $-169.42$ | $-148.90$ | $-141.17$ |
| $\mathcal{I}$-VAE (no TCL) | $-692.58$ | $-173.47$ | $-155.47$ | $-144.20$ | $-127.79$ |
| $\mathcal{I}$-VAE | $-657.39$ | $-159.04$ | $-147.93$ | $-136.54$ | $-125.45$ |

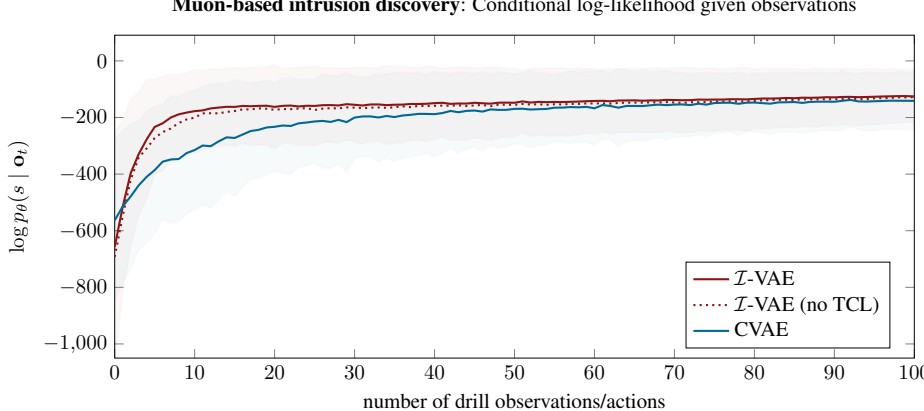

Figure 14: Conditional log-likelihood for the muon problem using the heuristic policy.

samples $k = 900$ state particles from the prior training data to represent its belief—using the same training states as the CVAE and $\mathcal{I}$-VAE models, given those states are the only examples each model has access to. Sampling the finite state particles results in the inability of the particle filter to generate new states that are consistent with the observations but may not directly exist in the finite prior data. This highlights the usefulness of learning generative models to act as approximate belief updaters, noting that we omit the particle filter results for the heuristic and oracle as they are prohibitively expensive given that the particle filter is not inherently parallelizable (further emphasizing the claim that batched belief updating is important).

Because the $\mathcal{I}$-VAE model explicitly learns parameters $\mu_\mathbf{o}$ and $\log \sigma_\mathbf{o}^2$ for the conditional prior $p_\psi(z \mid \mathbf{o})$, instead of a unit Gaussian, we can easily visualize the latent space using the t-SNE dimensionality reduction technique (Platzer, 2013). Visualized in Figure 16, we can inspect the latent space and qualitatively determine that it learns to cluster observations based on the economic volume of the reconstructed states (shown in the colors). This learned clustering is consistent over the training data in Figure 16a and the hold-out testing data in Figure 16b. The inlet plots show the average state reconstruction over 100 samples.

Finally, Figure 17 illustrates three example planning trajectories showing the belief mean over 100 sampled states (where drill action locations are marked in red) and we show the economic intrusion volume distribution. Quantitatively, in each of these three cases, we see that the heuristic planner using the $\mathcal{I}$-VAE model converges close to the true intrusion volume. These examples illustrate that the $\mathcal{I}$-VAE model accurately constructs a belief that converges, in shape, to the true state. The figure shows the mean belief and volume distributions

Table 2: Muon problem: Planning metrics comparing various policies and belief updaters.

| Method | Grid (horiz.) Accuracy ↑ # Actions ↓ | Grid (vert.) Accuracy ↑ # Actions ↓ | Random Accuracy ↑ # Actions ↓ | Heuristic Accuracy ↑ # Actions ↓ | Oracle Accuracy ↑ # Actions ↓ |
|---|---|---|---|---|---|
| Particle filter[†] | $0.75 \pm 0.03$ $89.00 \pm 1.03$ | $0.65 \pm 0.03$ $76.17 \pm 1.13$ | $0.67 \pm 0.03$ $76.52 \pm 1.02$ | — — | — — |
| CVAE | $0.92 \pm 0.02$ $27.38 \pm 1.79$ | $0.925 \pm 0.02$ $35.40 \pm 1.80$ | $0.925 \pm 0.02$ $24.36 \pm 2.05$ | $0.925 \pm 0.02$ $20.65 \pm 1.84$ | $0.935 \pm 0.02$ $21.86 \pm 1.85$ |
| $\mathcal{I}$-VAE (no TCL) | $0.925 \pm 0.02$ $28.85 \pm 1.24$ | $\mathbf{0.955 \pm 0.01}$ $34.02 \pm 1.53$ | $0.935 \pm 0.02$ $\mathbf{18.78 \pm 1.47}$ | $0.95 \pm 0.02$ $\mathbf{7.46 \pm 0.97}$ | $0.975 \pm 0.01$ $\mathbf{9.62 \pm 1.45}$ |
| $\mathcal{I}$-VAE | $0.925 \pm 0.02$ $28.45 \pm 1.31$ | $\mathbf{0.955 \pm 0.01}$ $\mathbf{31.11 \pm 1.39}$ | $\mathbf{0.95 \pm 0.02}$ $14.53 \pm 0.90$ | $\mathbf{0.96 \pm 0.01}$ $\mathbf{6.88 \pm 0.85}$ | $\mathbf{0.99 \pm 0.01}$ $\mathbf{8.15 \pm 1.22}$ |

[†] Heuristic and oracle results were omitted because the particle filter does not inherently support parallelism.

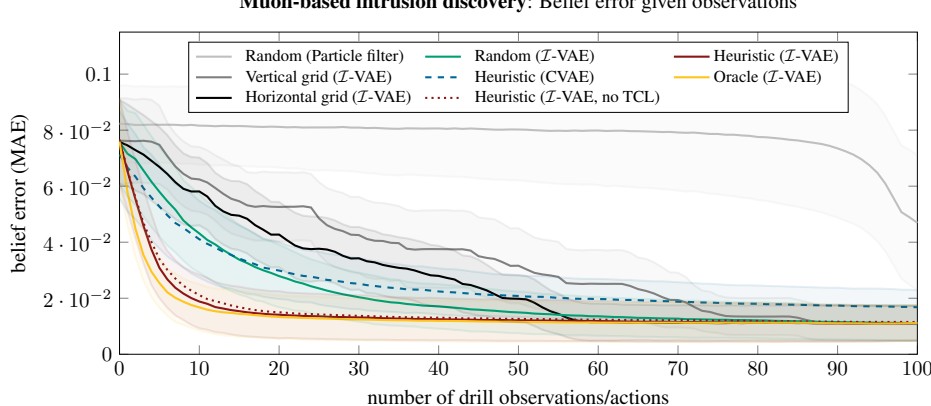

Figure 15: Belief error relative to holdout test states for different policies and updaters.

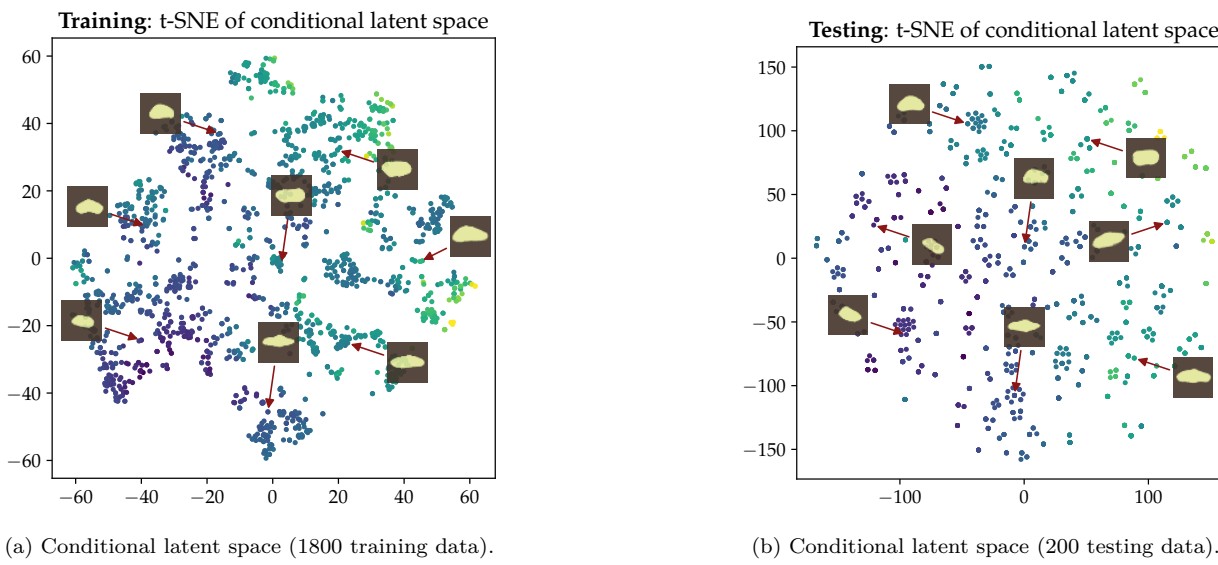

(a) Conditional latent space (1800 training data).

(b) Conditional latent space (200 testing data).

Figure 16: The conditional latent space $p_\psi(z \mid \mathbf{o})$, colored by intrusion volume.

after a fixed number of actions, as well as the final belief and volume distribution when all 100 drill actions are taken, where the blue lines indicate the true intrusion volume and the dashed red lines indicate the estimated mean from the belief.

# 7   Conclusions

In this work, we introduced the *inversion variational autoencoder* ($\mathcal{I}$-VAE) model that learns a latent distribution over states and a matching latent distribution over partial observations. Given our sequential decision making setting, we further refine the model during a fine-tuning step to maximize the mutual information between latent variables along the same observation trajectories. To improve efficiency in planning, we further formalized a batched extension to Markov decision processes and belief-state MDPs and introduced the *batched belief-state MDP* (**BB**-MDP). Requirements on a parallelized state transition function and reward function were necessary to adapt standard MDPs to the batched setting. For POMDPs, in addition to the requirements stated for MDPs, we also required parallelized observation functions and belief updaters.

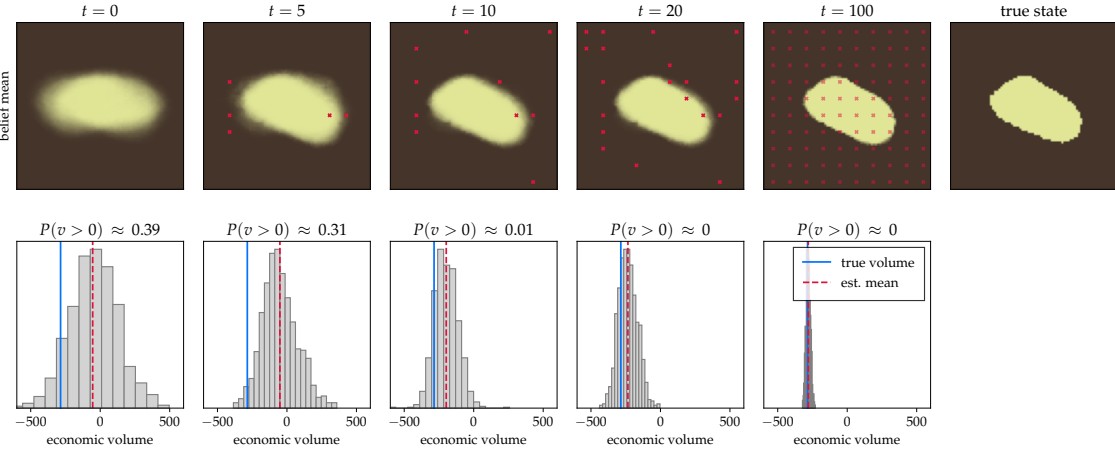

(a) Intrusion planning where the true volume is *below* the economic threshold.

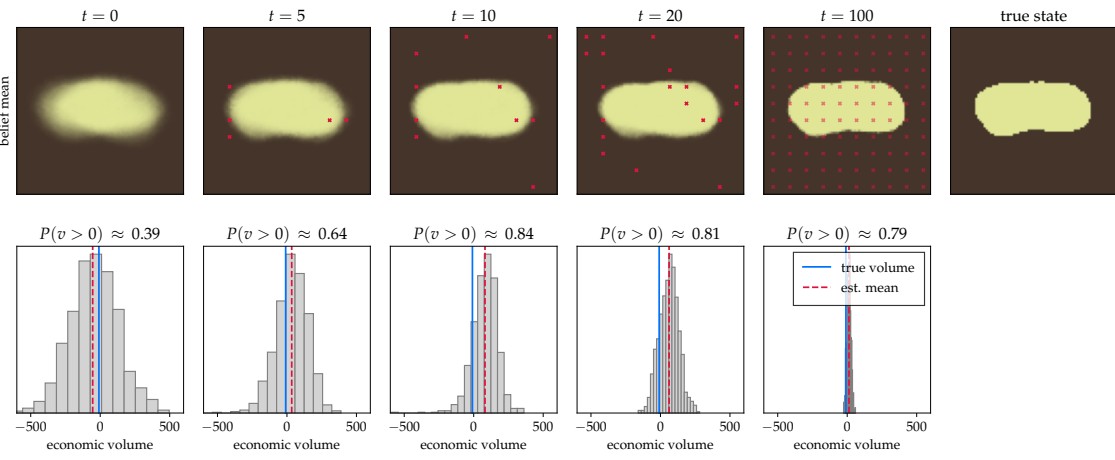

(b) Intrusion planning where the true volume is *near* the economic threshold.

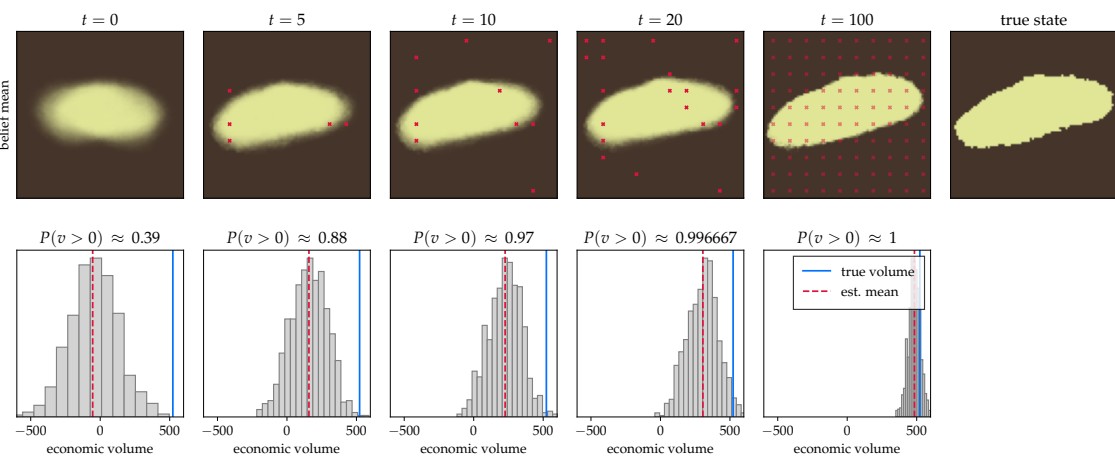

(c) Intrusion planning where the true volume is *above* the economic threshold.

Figure 17: Beliefs for the muon-based intrusion discovery problem using the $\mathcal{I}$-VAE model.

We showed that the $\mathcal{I}$-VAE model works well in cases where the observations are masked versions of the state and when the observations are indirectly related to the state through a forward model. We demonstrated the model performance when used as a surrogate belief updater in a muon-based intrusion discovery problem, and showed that an information-based heuristic policy works well in this setting. Insights from this work highlight that in problems with limited data, e.g., geological problems, using generative models allow for the creation of unseen states that are consistent with both the prior and the conditioning observations.

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

# A  Appendix

## A.1  KL-Divergence Derivation

In practice, the latent distributions over $\mathbf{z}$ are modeled as multivariate Gaussian distributions. Given the probability density function of a multivariate Gaussian with diagonal covariance:

$$\mathcal{N}\Big(\mathbf{z}; \boldsymbol{\mu}, \mathrm{diag}\left(\boldsymbol{\sigma}^2\right)\Big) = \prod_{i=1}^{|\mathbf{z}|} \frac{1}{\sqrt{2\pi\sigma_i^2}} \exp\left(-\frac{(z_i - \mu_i)^2}{2\sigma_i^2}\right) \tag{47}$$

we analytically derive the KL-divergence between the latent state distribution $q_\phi(\mathbf{z} \mid s)$ and the latent observation distribution $p_\psi(\mathbf{z} \mid \mathbf{o})$ (where $(\boldsymbol{\mu}_s, \boldsymbol{\sigma}_s)$ and $(\boldsymbol{\mu_o}, \boldsymbol{\sigma_o})$ are the parameters of their respective distributions) as:

$$\text{KL}\left(q_\phi(\mathbf{z}\,|\,s)\,\|\,p_\psi(\mathbf{z}\mid\mathbf{o})\right) \tag{48}$$

$$= \int q_\phi(\mathbf{z}\mid s)\log\left(\frac{q_\phi(\mathbf{z}\mid s)}{p_\psi(\mathbf{z}\mid\mathbf{o})}\right)\mathrm{d}\mathbf{z} \tag{49}$$

$$= \int q_\phi(\mathbf{z}\mid s)\log\left(\prod_{i=1}^{|\mathbf{z}|}\frac{\left(2\pi\sigma_{s,i}^2\right)^{-1/2}\exp\left(-(z_i-\mu_{s,i})^2/(2\sigma_{s,i}^2)\right)}{\left(2\pi\sigma_{\mathbf{o},i}^2\right)^{-1/2}\exp\left(-(z_i-\mu_{\mathbf{o},i})^2/(2\sigma_{\mathbf{o},i}^2)\right)}\right)\mathrm{d}\mathbf{z} \tag{50}$$

$$= \int q_\phi(z\mid s)\sum_{i=1}^{|\mathbf{z}|}\left(\log\left(\frac{\sigma_{\mathbf{o},i}}{\sigma_{s,i}}\right)-\frac{(z_i-\mu_{\mathbf{o},i})^2}{2\sigma_{\mathbf{o},i}^2}+\frac{(z_i-\mu_{s,i})^2}{2\sigma_{s,i}^2}\right)\mathrm{d}\mathbf{z} \tag{51}$$

$$= \sum_{i=1}^{|\mathbf{z}|}\int q_\phi(z_i\mid s)\left(\log\left(\frac{\sigma_{\mathbf{o},i}}{\sigma_{s,i}}\right)-\frac{(z_i-\mu_{\mathbf{o},i})^2}{2\sigma_{\mathbf{o},i}^2}+\frac{(z_i-\mu_{s,i})^2}{2\sigma_{s,i}^2}\right)\mathrm{d}\mathbf{z} \tag{52}$$

$$= \sum_{i=1}^{|\mathbf{z}|}\left(\frac{1}{2}\left(\log\sigma_{\mathbf{o},i}^2-\log\sigma_{s,i}^2\right)-\frac{1}{2}+\frac{1}{2\sigma_{\mathbf{o},i}^2}\left(\sigma_{s,i}^2+(\mu_{s,i}-\mu_{\mathbf{o},i})^2\right)\right) \tag{53}$$

$$= \frac{1}{2}\sum_{i=1}^{|\mathbf{z}|}\left(\log\sigma_{\mathbf{o},i}^2-\log\sigma_{s,i}^2+\frac{\sigma_{s,i}^2+(\mu_{s,i}-\mu_{\mathbf{o},i})^2}{\sigma_{\mathbf{o},i}^2}-1\right) \tag{54}$$

More complex latent distributions could instead be used and have been extensively studied in the literature; such as Gaussian mixture models (Dilokthanakul et al., 2016; Tomczak & Welling, 2018), the Gumbel-Softmax distribution for categorical latents (Jang et al., 2017), normalizing flows (Rezende & Mohamed, 2015), inverse autoregressive flows (Kingma et al., 2016), and diffusion models (Kingma et al., 2021).

## A.2 Batched Planning Problem Formulation

We here define the *batched Markov decision process* (**B**-MDP) as the tuple $\langle\mathcal{S}^m,\mathcal{A}^m,\mathbf{T},\widehat{R},\gamma\rangle$, where the batched state transition function $\mathbf{s}'\sim\mathbf{T}(\cdot\mid\mathbf{s},\mathbf{a})$ is defined as $\mathbf{T}:\mathcal{S}^m\times\mathcal{A}^m\to\mathcal{S}^m$, and the set of next states is:

$$\mathbf{s}'=\left(s_1',\ldots,s_m'\right)\quad\text{where}\quad s_i'\sim T(\cdot\mid s_i,a_i)\quad\text{for each}\quad(s_i,a_i)\in(\mathbf{s},\mathbf{a}) \tag{55}$$

The batched state space $\mathcal{S}^m$ is the space of ordered $m$-tuples of states, with $\mathbf{s}\in\mathcal{S}^m$ denoting batched states and $\mathbf{a}\in\mathcal{A}^m$ denoting batched actions. The transition function $\mathbf{T}$ can be implemented using a parallelized (e.g., vectorized) form of the standard MDP dynamics. In practice, this can be done either through a parallelized simulator, vectorizing the transition function, or a learned neural network surrogate of the transition function for easy GPU vectorization. The batched state-based reward function is the expectation of the underlying reward function over the $m$ batched states and actions and is defined as:

$$\widehat{R}(\mathbf{s},\mathbf{a})=\frac{1}{m}\sum_{i=1}^m R(s_i,a_i) \tag{56}$$

### A.2.1 Batched Value Decomposition and Optimality

A batched MDP preserves the optimality of the value function in the underlying MDP. The key observation is that, under the batched dynamics $\mathbf{T}$ with batched reward $\widehat{R}(\mathbf{s},\mathbf{a})$, the process factorizes across components. This factorization induces a natural decomposition of the batched value function. Therefore, the batched Bellman equations decouple, and for any stationary policy $\pi$ with batched extension $\boldsymbol{\pi}$ the batched value is simply the average of the individual value functions. In particular, if $\pi^*$ is optimal for the underlying MDP, then for every batch state $\mathbf{s}=(s_1,\ldots,s_m)$, applying $\pi^*$ independently to each component is optimal in the batched MDP. We show in Theorem 1 in A.4 that the batched value function under the optimal batched policy $\boldsymbol{\pi}^*$ is equivalent to the average of the optimal values of the underlying MDP. Optimality is therefore preserved under the batched extension.

### A.2.2 Batched Optimal Policy

This section details how we select optimal independent actions based on the batched value function $\widehat{V}$. Using the definition of the state-action value (i.e., the $Q$-value) from the underlying MDP, we get:

$$V^*(s) = \max_{a \in \mathcal{A}} Q^*(s, a) \tag{57}$$

Using the results from Theorem 1 and applying the Bellman operator, we get the following:

$$\widehat{V}^*(\mathbf{s}) = \frac{1}{m} \sum_{i=1}^{m} \max_{a \in \mathcal{A}} \left( R(s_i, a) + \gamma \sum_{s'} T(s' \mid s_i, a) \max_{a' \in \mathcal{A}} Q^*(s', a') \right) \tag{58}$$

Similarly, we can derive the optimal batched state-action $Q$-value function:

$$\widehat{Q}^*(\mathbf{s}, a) = \frac{1}{m} \sum_{i=1}^{m} \left( R(s_i, a) + \gamma \sum_{s'} T(s' \mid s_i, a) \max_{a' \in \mathcal{A}} Q^*(s', a') \right) \tag{59}$$

We define the optimal batched policy under independent actions as the vector of per-state optimal decisions which captures uncertainty in the per-state dynamics:

$$\boldsymbol{\pi}^*(\mathbf{s}) = \left( \arg\max_{a \in \mathcal{A}} Q^*(s_i, a) \mid s_i \in \mathbf{s} \right) \tag{60}$$

This policy defines a batch of locally optimal actions that achieve the maximum value for each individual $s_i$ in the batched planning model.

While $\boldsymbol{\pi}^*(\mathbf{s})$ is not a valid execution policy in environments where only one action may be taken, it serves as a useful intermediate representation for analyzing the structure of the batched value function and for planning under independent state samples. The next section discusses how the batched planning model is used in the true single-state environment.

### A.3 Single-State Batched Planning

Although the true environment is modeled as a standard MDP $\langle \mathcal{S}, \mathcal{A}, T, R, \gamma \rangle$, in which the agent is only in a single state $s \in \mathcal{S}$ at each timestep and must select a single action $a \in \mathcal{A}$, we formulated the batched planning framework in A.2 to enable robust and efficient evaluation of potential future outcomes. This section describes how we use the batched state representation for planning while satisfying the requirement that the agent selects only one action to execute in the true environment.

### A.3.1 Constructing the Batched Planning Model

To construct a batched planning representation, we replicate the current state $s$ a total of $m$ times to form a batched state:

$$\mathbf{s} = (s, \ldots, s) \in \mathcal{S}^m \tag{61}$$

This batched state is used internally for planning purposes, such as simulating forward transitions under stochastic dynamics, evaluating expected returns, or propagating value estimates in parallel.

Transitions and rewards in the batched model follow the independent structure introduced in our formulation of the batched MDP. That is, for a given action $a \in \mathcal{A}$ and duplicated to get $\mathbf{a} = (a, \ldots, a) \in \mathcal{A}^m$, each element of the batch evolves independently as $s_i' \sim T(\cdot \mid s_i, a_i)$ for each $s_i = s \in \mathbf{s}$ and $a_i = a \in \mathbf{a}$. The rewards are averaged across the batch as $\widehat{R}(\mathbf{s}, \mathbf{a}) = \frac{1}{m} \sum_{i=1}^{m} R(s_i, a_i) = R(s, a)$ since all $s_i$ and $a_i$ are identical.

### A.3.2 Selecting a Single Action

Although planning is performed over the batched state $\mathbf{s}$ and subsequent batched future states $\mathbf{s}'$, the final policy must ultimately select a single action to execute in the true environment. We therefore use the batched

optimal $Q$-value function to express the single state-action $Q$-value function as:

$$\widehat{Q}^*(\mathbf{s}, a) = \frac{1}{m} \sum_{i=1}^{m} Q^*(s_i, a) = Q^*(s, a) \tag{62}$$

with $s_i = s$ for all $i$. This is only true in the degenerate case where all batch elements are identical. The expansion of $\widehat{Q}^*$ is useful for the general batched planning setting defined in A.2, but redundant when $\mathbf{s} = (s)_{i=1}^{m}$. The resulting policy is then given by:

$$\pi^*(s) = \underset{a \in \mathcal{A}}{\arg\max}\, \widehat{Q}^*(\mathbf{s}, a) \tag{63}$$

where the recursive definition of $\widehat{Q}^*$ in Equation (59) expands each $s \in \mathbf{s}$ based on the transition dynamics. This policy selects the action that maximizes the expected value across stochastic state transitions from the single known current state. Importantly, this formulation allows us to use the computational benefits of vectorized or parallelized transitions while maintaining full compatibility with a standard single-agent MDP interface.

### A.3.3 Interpretation

The batched formulation can be interpreted as a Monte Carlo approximation of the expected return under stochastic transitions. Specifically, given a single state $s$ and candidate action $a$, we simulate $m$ stochastic next states $s_i \sim T(\cdot \mid s, a)$ and compute:

$$\widehat{Q}^*(\mathbf{s}, a) \approx R(s, a) + \gamma \frac{1}{m} \sum_{i=1}^{m} V^*(s_i') \tag{64}$$

which approximates the Bellman backup:

$$Q^*(s, a) = R(s, a) + \gamma \mathbb{E}_{s' \sim T(\cdot|s,a)} \big[ V^*(s') \big] \tag{65}$$

for $s \in \mathbf{s}$. By evaluating this expectation over $m$ simulated next states, the policy can make informed action choices that account for environmental uncertainty without explicitly modeling a belief state. This formulation also supports integration with planning algorithms such as Monte Carlo tree search (MCTS) (Grill et al., 2020; Cazenave, 2022), rollout-based policy evaluation (Bertsekas, 2021), or batched value iteration (Ernst et al., 2005).

### A.4 Batched Value Decomposition and Optimality Proof

**Theorem 1 (Batched value decomposition)** *Let $\pi : \mathcal{S} \to \mathcal{A}$ be any stationary policy over the underlying MDP, and let $\boldsymbol{\pi} : \mathcal{S}^m \to \mathcal{A}^m$ be its batched extension applied independently to each element of $\mathbf{s} \in \mathcal{S}^m$ where $\boldsymbol{\pi}(\mathbf{s}) = \big(\pi(s_1), \ldots, \pi(s_m)\big)$. Then the batched value function satisfies:*

$$\widehat{V}^{\boldsymbol{\pi}}(\mathbf{s}) = \frac{1}{m} \sum_{i=1}^{m} V^{\pi}(s_i) \tag{66}$$

*and in particular, if $\pi^*$ is optimal for the underlying MDP, then:*

$$\widehat{V}^{\boldsymbol{\pi}^*}(\mathbf{s}) = \frac{1}{m} \sum_{i=1}^{m} V^*(s_i) \tag{67}$$

*That is, the batched value function preserves optimality in expectation.*

**Proof** The batched value function under policy $\boldsymbol{\pi}$ can be trivially derived following the linearity of expectation:

$$\widehat{V}^{\boldsymbol{\pi}}(\mathbf{s}) = \mathbb{E}\left[\sum_{t=0}^{\infty}\gamma^t\widehat{R}\Big(\mathbf{s}^{(t)},\boldsymbol{\pi}\big(\mathbf{s}^{(t)}\big)\Big)\,\bigg|\,\mathbf{s}^{(0)}=\mathbf{s}\right] \tag{68}$$

$$= \mathbb{E}\left[\sum_{t=0}^{\infty}\gamma^t\left(\frac{1}{m}\sum_{i=1}^{m}R\Big(s_i^{(t)},\pi\big(s_i^{(t)}\big)\Big)\right)\,\bigg|\,(s_1^{(0)},\dots,s_m^{(0)})=\mathbf{s}\right] \tag{69}$$

$$= \frac{1}{m}\sum_{i=1}^{m}\mathbb{E}\left[\sum_{t=0}^{\infty}\gamma^t R\Big(s_i^{(t)},\pi\big(s_i^{(t)}\big)\Big)\,\bigg|\,s_i^{(0)}=s_i\right] \tag{70}$$

$$= \frac{1}{m}\sum_{i=1}^{m}V^{\pi}(s_i) \tag{71}$$

If $\pi^*$ is an optimal policy for the underlying MDP, then $V^{\pi^*}(s_i) = V^*(s_i)$ for all $s_i \in \mathcal{S}$. Substituting this into the batched value expression gives:

$$\widehat{V}^{\boldsymbol{\pi}^*}(\mathbf{s}) = \frac{1}{m}\sum_{i=1}^{m}V^*(s_i) \tag{72}$$

Thus, the batched value function under the optimal batched policy $\boldsymbol{\pi}^*$ is equivalent to the average of the optimal values of the underlying MDP. Therefore, optimality is preserved under the batched extension. ∎

### A.5    Belief Expected Entropy Reduction Proof

**Theorem 2 (Expected entropy reduction and KL-divergence equivalence)** *Let $b$ be a prior belief over state space $\mathcal{S}$. Suppose we draw $m$ posterior beliefs $(b'_1,\dots,b'_m) = \mathbf{b}'$ by sampling $b'_i \sim T_b(\cdot \mid b, a_i)$ following Equations* (28) *to* (31)*, then the expected reduction in entropy is equivalent to the expected KL-divergence between $\mathbf{b}'$ and $b$, namely:*

$$\mathcal{H}(b) - \mathbb{E}_i\big[\mathcal{H}(b'_i)\big] = \mathbb{E}_i\big[\,\mathrm{KL}(b'_i \,\|\, b)\big] \quad where \quad b'_i \in \mathbf{b}' \tag{73}$$

*This results in an optimal one-step information gathering policy that maximizes Bayesian information gain. The outer expectation over posterior beliefs can be approximated using the Monte Carlo average over the $m$ sampled updated beliefs:*

$$\mathcal{H}(b) - \mathbb{E}_i\left[\mathcal{H}(b'_i)\right] \approx \mathcal{H}(b) - \frac{1}{m}\sum_{i=1}^{m}\mathcal{H}(b'_i). \tag{74}$$

*Under standard Monte Carlo assumptions, this empirical average converges to the expected posterior entropy as $m \to \infty$. The entropy $\mathcal{H}(b'_i)$ may be evaluated exactly or estimated from the chosen belief representation.*

**Proof** We show that the expected entropy reduction of the belief is exactly the expected KL-divergence between each posterior and the prior, justifying the use of this measurement when selecting actions to maximize expected information gain. For each posterior belief $b'_i$, the KL-divergence can be expressed as:

$$\mathrm{KL}(b'_i \,\|\, b) = \int b'_i(s)\log b'_i(s)\,\mathrm{d}s - \int b'_i(s)\log b(s)\,\mathrm{d}s \tag{75}$$

$$= -\mathcal{H}(b'_i) - \int b'_i(s)\log b(s)\,\mathrm{d}s \tag{76}$$

Rearranging and adding $\mathcal{H}(b)$ to both sides, we get:

$$\mathcal{H}(b) - \mathcal{H}(b'_i) = \mathrm{KL}(b'_i \,\|\, b) + \int b'_i(s)\log b(s)\,\mathrm{d}s + \mathcal{H}(b) \tag{77}$$

Because $b'_i$ is generated by first sampling $s \sim b$, followed by $s' \sim T(\cdot \mid s, a)$, $o \sim O(\cdot \mid a, s')$, and updated using $b' = \text{UPDATE}(b, a, o)$, and by the law of total expectation, we get:

$$\mathbb{E}_i \left[ \int b'_i(s) \log b(s) \, \mathrm{d}s \right] = \mathop{\mathbb{E}}_{s \sim b} \left[ \mathop{\mathbb{E}}_{(s', o \mid s)} \left[ \mathbb{E}_{b'_i} \left[ \log b(s) \right] \right] \right] \tag{78}$$

$$= \mathbb{E}_s \left[ \log b(s) \right] \tag{79}$$

$$= \int b(s) \log b(s) \, \mathrm{d}s \tag{80}$$

$$= -\mathcal{H}(b) \tag{81}$$

Now using this result in the expanded definition of expected entropy reduction, we get:

$$\mathcal{H}(b) - \mathbb{E}_i \left[ \mathcal{H}(b'_i) \right] = \mathbb{E}_i \left[ \mathcal{H}(b) - \mathcal{H}(b'_i) \right] \tag{82}$$

$$= \mathbb{E}_i \left[ \text{KL}(b'_i \,\|\, b) + \underbrace{\int b'_i(s) \log b(s) \, \mathrm{d}s}_{-\mathcal{H}(b)} + \mathcal{H}(b) \right] \tag{83}$$

$$= \mathbb{E}_i \left[ \text{KL}(b'_i \,\|\, b) \right] \tag{84}$$

Therefore, we reach our desired equivalence of:

$$\mathcal{H}(b) - \mathbb{E}_i \left[ \mathcal{H}(b'_i) \right] = \mathbb{E}_i \left[ \text{KL}(b'_i \,\|\, b) \right] \tag{85}$$

∎

## A.6 Prior Subsurface Intrusion Data Generation

The muon-based intrusion discovery case study was derived from critical mineral exploration in Bulloo Downs, Queensland, Australia. The initial data in this area is the observed gravity geophysical data showing the anomaly of a denser geological body existing in the subsurface. This denser geobody is hypothesized to be a mafic or ultramafic magmatic intrusion that hosts Ni–Cu–PGE sulfide mineralization (Barnes et al., 2016), as shown in Figure 18. Following the levelset inversion approach and implementation from Wang et al. (2023), a stochastic geophysical inversion using levelsets approach was performed with the observed gravity data and geological hypothesis (3D geometry of the magmatic intrusion) as the input. To do so, we translate the observed gravity data and 3D intrusion geometry into a loss function. We then apply Markov Chain Monte Carlo (MCMC) sampling to minimize this loss function by stochastically generating 3D intrusive bodies that can predict the observed gravity data while conforming the hypothesized geological shape. The inversion results obtained a total of 2000 intrusive geological body samples that match the observed gravity data, with a hypothesized geometry from (Barnes et al., 2016). The 2000 samples are used to train our muon tomography $\mathcal{I}$-VAE model, splitting the data into 1800 training and 200 testing points.

## A.7 Additional MNIST Benchmark Analysis

Figures 19 and 20 provide additional qualitative analysis for the MNIST benchmark. The figures show representative examples of taking random pixel observation actions from 0% to 100% coverage for the $\mathcal{I}$-VAE and CVAE. We plot the mean belief over 1000 samples and several generated state samples over time. These results also indicate that the $\mathcal{I}$-VAE not only learns better and more diverse digit reconstructions, as evident by the more visually consistent predictions, but also converges to the correct state in fewer observations (about 10% in Figure 19 and about 20% in Figure 20). We also include the aggregate classification probabilities from a learned classification MLP (Imambi et al., 2021) in the bottom row, further indicating the confidence in the prediction when aggregating over the 1000 sampled states. In the classification plots, the shaded bars represent the correct classification label. In both models, when the observation coverage is high, the reconstruction recovers the true state.

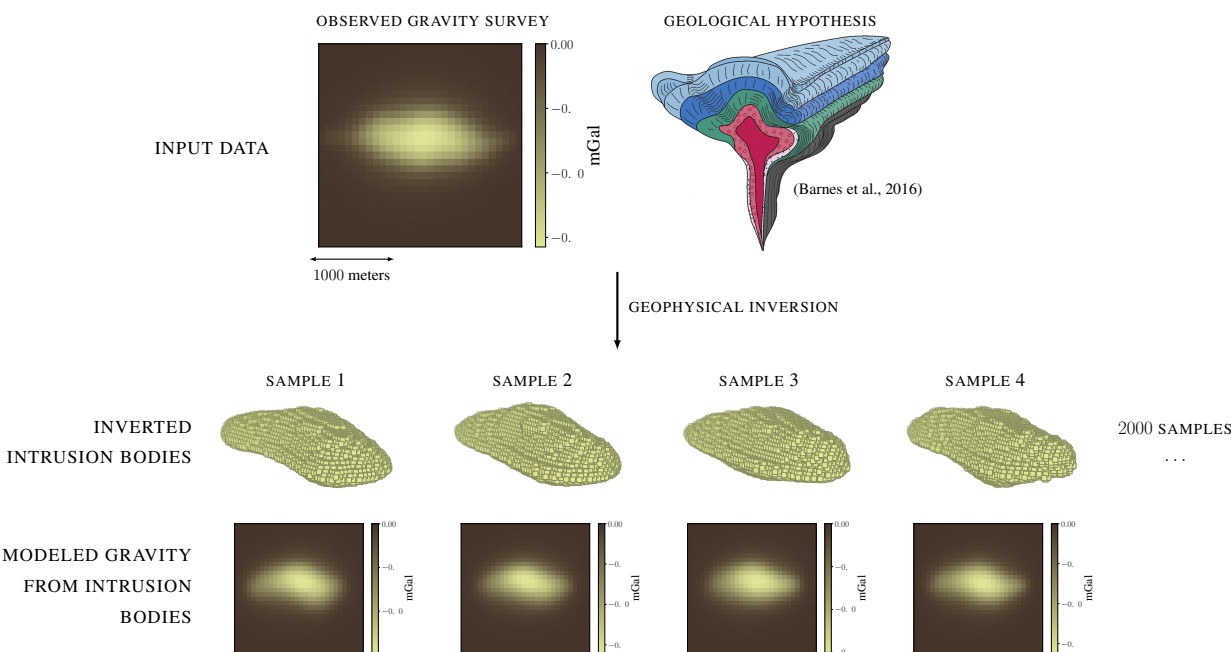

Figure 18: Prior muon data generation from observed gravity and a geological hypothesis.

Table 3: Inference time and sliced Wasserstein distance (SWD), representing posterior calibration error, on the Gaussian-mixture inverse problem. Timing is median milliseconds per batched posterior-sampling update and uncertainty is one standard error across seeds.

| Method | ms/update | Rel. time | SWD 0% | SWD 25% | SWD 50% | SWD 100% |
|---|---|---|---|---|---|---|
| $\mathcal{I}$-VAE | 0.303 ± 0.004 | 1.0× | 0.116 ± 0.002 | 0.110 ± 0.002 | 0.103 ± 0.001 | 0.085 ± 0.002 |
| $\mathcal{I}$-VAE (no TCL) | 0.303 ± 0.003 | 1.0× | 0.118 ± 0.004 | 0.116 ± 0.003 | 0.105 ± 0.001 | 0.086 ± 0.003 |
| CVAE | 0.249 ± 0.013 | 0.8× | 0.120 ± 0.004 | 0.114 ± 0.001 | 0.110 ± 0.001 | 0.089 ± 0.002 |
| Diffusion-5 | 2.868 ± 0.016 | 9.5× | 0.182 ± 0.010 | 0.153 ± 0.003 | 0.134 ± 0.001 | 0.105 ± 0.002 |
| Diffusion-10 | 5.778 ± 0.004 | 19.1× | 0.145 ± 0.004 | 0.146 ± 0.003 | 0.135 ± 0.001 | 0.112 ± 0.004 |

## A.8 Gaussian-Mixture Prior Experiment

As documented in Table 3 and Figure 21, we introduce a four-component Gaussian-mixture inverse problem with noisy observations and a closed-form posterior. This benchmark enables evaluation of posterior accuracy directly without relying on a downstream planning task. We compare $\mathcal{I}$-VAE, $\mathcal{I}$-VAE (no TCL), CVAE, and a conditional diffusion baseline against the analytical posterior using sliced Wasserstein distance (SWD), and report inference time for each. All methods use the same training data distribution, batch size, and base update budget. The results show that $\mathcal{I}$-VAE achieves the lowest SWD across selected observation coverage levels while retaining single-pass inference. In contrast, diffusion sampling incurs a substantial inference-time cost: Diffusion-5 is approximately 9.5× slower than $\mathcal{I}$-VAE, and Diffusion-10 is approximately 19.1× slower. Increasing the number of denoising steps improves diffusion accuracy in the low-observation regime, but the improvement does not close the gap to $\mathcal{I}$-VAE and comes at roughly double the inference time. Overall, the Gaussian-mixture benchmark results suggest that $\mathcal{I}$-VAE produces posterior samples that are both computationally efficient and closely aligned with the analytical posterior.

## A.9 TCL Ablation

We ablate TCL weighting parameter $\alpha$ in the MNIST benchmark as shown in Figure 22. TCL fine-tuning improves conditional log-likelihood over the no-TCL $\mathcal{I}$-VAE baseline across a broad range of $\alpha$, with the largest gains occurring in the 0-10% low-observation regime. With no observations, TCL can only affect the

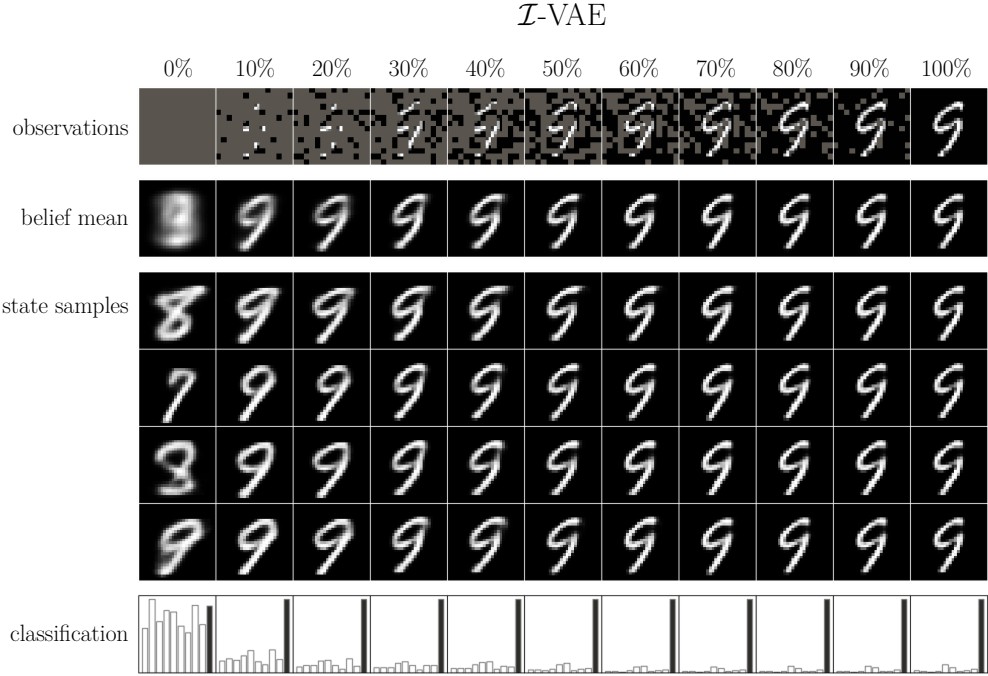

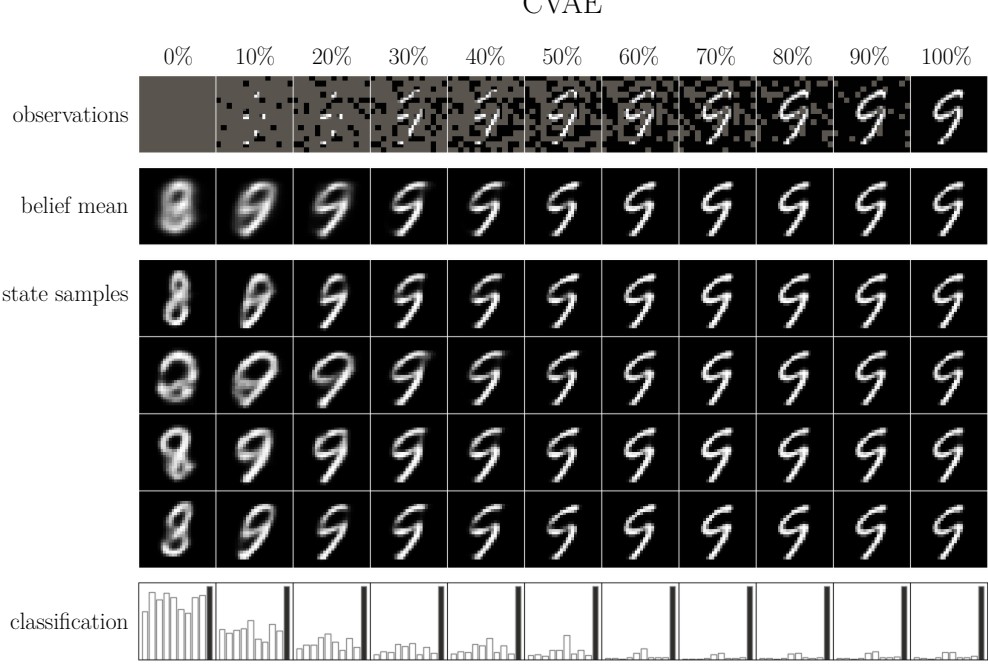

Figure 19: MNIST example observations over time for the $\mathcal{I}$-VAE and CVAE models.

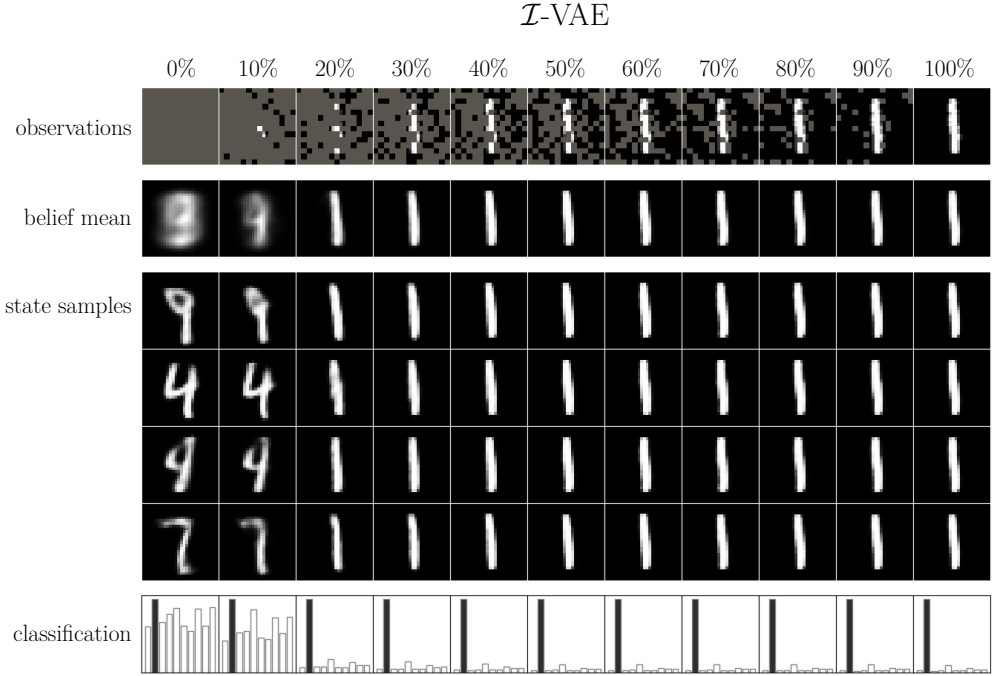

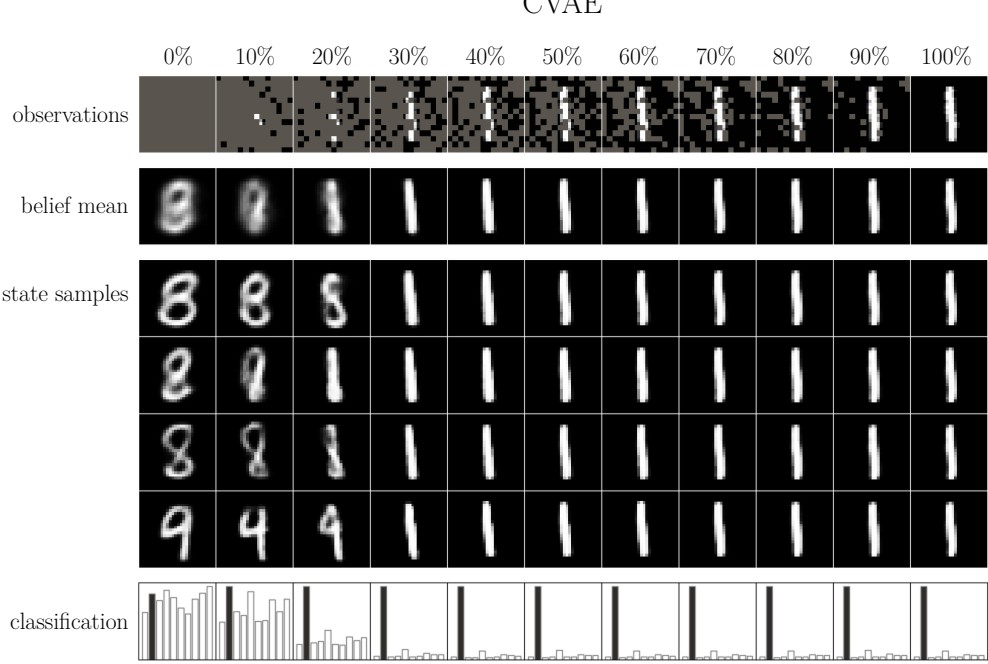

Figure 20: Additional MNIST observations over time for the $\mathcal{I}$-VAE and CVAE models.

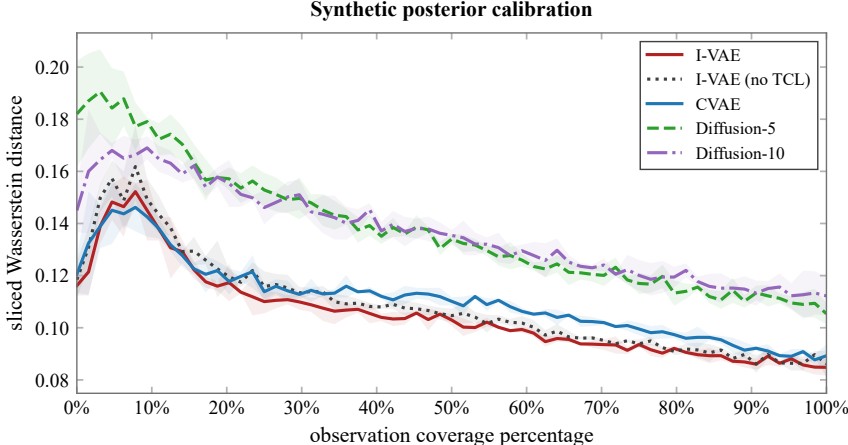

Figure 21: Gaussian-mixture posterior accuracy for autoencoder and diffusion methods.

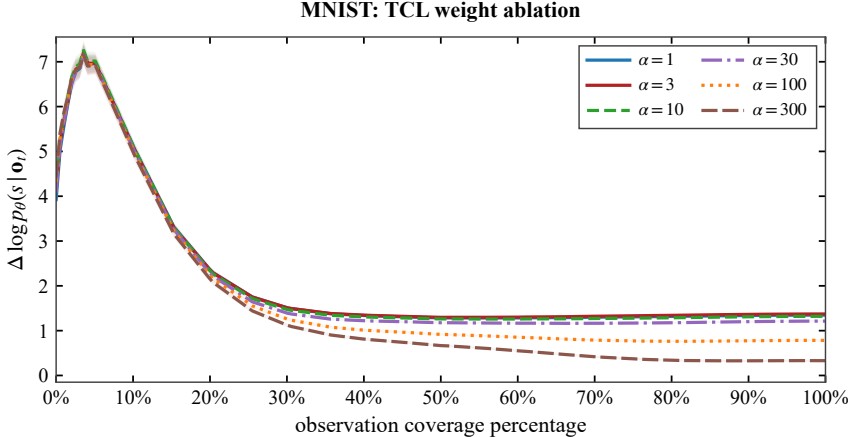

Figure 22: TCL weight $\alpha$ ablation on MNIST dataset.

unconditioned prior. A small number of observed patches provides enough information to identify coarse digit structure while leaving most pixels unobserved, making posterior alignment especially valuable. As coverage increases, the conditional inference problem becomes less ambiguous and the no-TCL model catches up, reducing the relative CLL gain. Ablation results indicate that the reported improvement is not sensitive to a narrow hyperparameter choice.

## A.10 Notation

Table 4: Notation used throughout the paper and their definitions.

| Symbol | Meaning |
|---|---|
| **States, observations, beliefs, and inversion** | |
| $\mathcal{S}$ | State space. |
| $s \in \mathcal{S}$ | Hidden state. In an EMDP, the state is static. |
| $s'$ | Next-state variable in a POMDP transition. In an EMDP, $s' = s$. |
| $\mathcal{A}$ | Action space. |
| $a \in \mathcal{A}$ | Action selected by the agent. |

| Symbol | Meaning |
|---|---|
| $\mathcal{O}$ | Observation space. |
| $o \in \mathcal{O}$ | A single observation. |
| $o_{1:t}$ | Observation history up to time $t$. |
| $\mathbf{o}$ | Shorthand for accumulated partial observations, typically $o_{1:t}$. |
| $a_{1:t}$ | Action history up to time $t$. |
| $T(s' \mid s, a)$ | State transition probability or density. |
| $O(o \mid a, s')$ | Observation model or likelihood in a POMDP. In an EMDP, written $O(o \mid a, s)$. |
| $R(s, a)$ | Immediate reward for taking action $a$ in state $s$. |
| $\gamma$ | Discount factor. |
| $\eta$ | Normalizing constant in the Bayesian belief update. |
| $f$ | Forward observation process mapping states and actions to observations. |
| $f^{-1}$ | Inverse map from partial observations to a state estimate or posterior state distribution. |
| $p(s \mid \mathbf{o})$ | Posterior distribution over hidden states conditioned on partial observations. |
| $\mathcal{B}$ | Belief space over $\mathcal{S}$. |
| $b \in \mathcal{B}$ | Belief state, represented in this work by posterior state particles. |
| $b'$ | Updated belief after incorporating an action and observation. |
| $b = (s_1, \ldots, s_k)$ | Particle belief with $k$ posterior state samples. |
| $s_i \sim p(\cdot \mid \mathbf{o})$ | State particle sampled from the posterior conditioned on observations. |
| $k$ | Number of state particles used to represent one belief in the **BB**-MDP. |
| $\text{UPDATE}(b, a, o)$ | Belief update operator. |
| $R_b(b, a)$ | Single-belief reward, $R_b(b, a) = \int b(s) R(s, a)\, \mathrm{d}s$. |
| $\mathcal{H}(b)$ | Entropy of belief $b$. |
| $\text{KL}(p\|q)$ | Kullback–Leibler divergence from $q$ to $p$. |
| $\widehat{\mathcal{H}}_{\text{marg}}(b)$ | Marginal particle entropy estimator used to score particle beliefs. |
| $\widehat{\mathcal{I}}_{\text{marg}}(a; b)$ | Empirical information score for action $a$. |

**Batched planning and BB-MDP notation**

| | |
|---|---|
| $m$ | Number of batched elements, e.g., batched states, belief branches, or rollout branches. |
| $n_A = |\mathcal{A}|$ | Number of candidate actions evaluated during planning. Using $n_A$ avoids overloading $n$, which is also used for dataset size. |
| $\mathcal{S}^m$ | Batched state space. |
| $\mathcal{A}^m$ | Batched action space. |
| $\mathcal{O}^m$ | Batched observation space. |
| $\mathcal{B}^m$ | Batched belief-state space. |
| $\mathbf{s} = (s_1, \ldots, s_m)$ | Batched state. |
| $\mathbf{a} = (a_1, \ldots, a_m)$ | Batched action vector. |
| $\mathbf{o} = (o_1, \ldots, o_m)$ | Batched observation vector. |
| $\mathbf{b} = (b_1, \ldots, b_m)$ | Batched belief state. |
| $\mathbf{T}$ | Vectorized state-transition function. |
| $\mathbf{O}$ | Vectorized observation function. |
| $\mathbf{T}_b$ | Batched belief-state transition function. |
| $\overrightarrow{\text{UPDATE}}$ | Vectorized belief update applied across batch elements. |
| $\widehat{R}(\mathbf{s}, \mathbf{a})$ | Batched state reward, $\widehat{R}(\mathbf{s}, \mathbf{a}) = \frac{1}{m} \sum_{i=1}^{m} R(s_i, a_i)$. |
| $\widehat{R}_b(\mathbf{b}, \mathbf{a})$ | Batched belief-state reward, $\widehat{R}_b(\mathbf{b}, \mathbf{a}) = \frac{1}{m} \sum_{i=1}^{m} R_b(b_i, a_i)$. |
| $V^\pi(b)$ | Belief-state value function under policy $\pi$. |
| $Q^*(b, a)$ | Optimal belief-state action-value function. |
| $\widehat{V}^{\boldsymbol{\pi}}(\mathbf{b})$ | Batched belief-state value function under batched policy $\boldsymbol{\pi}$. |
| $\widehat{Q}^*(\mathbf{b}, a)$ | Batched belief-state action-value function used for action selection. |
| $\pi$ | Policy selecting a single executable action. |
| $\boldsymbol{\pi}$ | Batched policy applied independently across batch elements. |

| Symbol | Meaning |
|---|---|
| $\pi^*$ | Optimal policy. |

**VAE, CVAE, and $\mathcal{I}$-VAE notation**

| Symbol | Meaning |
|---|---|
| $z$ | Latent variable. |
| $d$ | Latent dimension. |
| $z_s$ | Latent variable associated with the state encoder distribution. |
| $z_{\mathbf{o}}$ | Latent variable sampled from the observation-conditioned latent prior. |
| $h_{\mathbf{o}}$ | Encoded observation representation. |
| $\mathcal{E}_{\mathbf{o}}$ | Observation encoder. |
| $q_\phi(z \mid s)$ | $\mathcal{I}$-VAE state encoder or recognition model. |
| $q_\phi(z \mid s, \mathbf{o})$ | CVAE recognition model. |
| $p_\theta(z \mid \mathbf{o})$ | CVAE conditional prior. |
| $p_\psi(z \mid \mathbf{o})$ | $\mathcal{I}$-VAE observation-conditioned latent prior. |
| $p_\theta(s \mid z, \mathbf{o})$ | Decoder/generative model for reconstructing or sampling states. |
| $\hat{s}$ | Reconstructed or generated state sample. |
| $\mu_s, \log \sigma_s^2$ | Mean and log-variance parameters associated with $q_\phi(z \mid s)$. |
| $\mu_{\mathbf{o}}, \log \sigma_{\mathbf{o}}^2$ | Mean and log-variance parameters associated with $p_\psi(z \mid \mathbf{o})$. |
| $\phi$ | Parameters of the recognition model. |
| $\psi$ | Parameters of the observation-conditioned latent prior. |
| $\theta$ | Parameters of the decoder/generative model. |
| $\mathcal{L}_{\text{VAE}}$ | VAE training loss. |
| $\mathcal{L}_{\text{CVAE}}$ | CVAE training loss. |
| $\mathcal{L}_{\mathcal{I}\text{-VAE}}$ | $\mathcal{I}$-VAE pretraining loss: reconstruction through $p_\theta(s \mid z, \mathbf{o})$ plus latent matching via $\text{KL}(q_\phi(z \mid s) \| p_\psi(z \mid \mathbf{o}))$. |

**Trajectory contrastive fine-tuning**

| Symbol | Meaning |
|---|---|
| $\mathcal{L}_{\text{TCL}}$ | Trajectory contrastive loss. |
| $\mathbf{o}_{1:T}$ | Observation trajectory over horizon $T$. |
| $\mathbf{o}_t$ | Partial observation at step $t$, equivalent to accumulated observations up to that step. |
| $z_t$ | Observation-inferred latent variable at step $t$. |
| $K$ | Lookahead window length in the TCL objective. |
| $\mathcal{I}(\cdot\,;\cdot)$ | Mutual information. |
| $\alpha$ | Weight on the trajectory contrastive loss during fine-tuning. |
| $\Pi_\omega$ | Projection network used in the contrastive objective. |
| $\omega$ | Projection network parameters. |
| $\tilde{z}$ | Projected latent representation, $\tilde{z} = \Pi_\omega(\mu_{\mathbf{o}_t})$. |
| $\mathbf{o}^+$ | Positive future observation from the same trajectory. |
| $\Omega^-$ | Set of negative observation sequences. |
| $\tau$ | Temperature parameter in the contrastive similarity function. |

