# OpenReview forum: "Partial Observation Inversion and Batched Belief-State Planning for Information Gathering POMDPs"
_TMLR — Decision pending for TMLR_

### Review · Reviewer_dYNv · 2026-04-03

**Summary Of Contributions:**

The authors propose the framework of an Inversion-VAE (I-VAE) and additionally a fine-tuning stage in the form of a contrastive loss for trajectories, termed TCL. The demonstrate their framework on the MNIST dataset and real world example in the context of mining, where the true latent state must be uncovered from a set of observations. Overall the mining taks is a very interesting and novel application.

**Audience:**

Yes

**Audience Explanation:**

The task itsself is of high relevance and interesting.

**Claims And Evidence:**

No

**Claims Explanation:**

On the positive side: The authors provide open source code to their and a set of experiments.
On the negative side: The approach itsself is theoretically questionable and confusing. More und the bullet point "requested changes".

**Requested Changes:**

(1-3) are major for recommending acceptance.

(1) When talking about VAEs, it is good practice to define the generative model
p(z_0), p(z_t| z_t-1, a_t), p(x_t|z_t), \pi(a_t) make the corresponding changes and clearly define all variables. Same goes for the posterior. I miss this part and it makes the paper hard to follow

(2) I am confused about the usage of "o", "s", and "z", It seems that "o" are the observations and the authors call sometimes "s" the latent, while "z" is actually the latent. There is also model-component "state-encoder" that takes the "true hidden state s" on page 16. Note that the hidden state is hidden, and you should not have access to it (the hidden state is "z". It seems to me that the the authors mix-up the usage of  states. This is commend is also inline with my point in (1)

(3) The authors sell going from q_psi(z | s, o) to q_psi(z | s) as a novelty. However, leaving out variables in the approx posterior is done regularly (see "Mind the Gap when Conditioning Amortised Inference in Sequential Latent-Variable Models" from Justin Bayer) and its actualy probable bad practice to omit variables on the conditioing side. Pls discuss this in detail, as there is no theoretical reason for this step.

(4) The TCL seems like a heuristic/balcony, as the VAE loss function is already proper. While the experiments show that the its beneficial, I miss a theoretical justification.

---

> ### Author Response · Authors · 2026-04-15
> **Rebuttal by Authors**
>
> We thank the reviewer for their consideration of our manuscript and for their assessment that “the task itself is of high relevance and interesting” and for noting the strength of the experiments and reproducibility. We believe the theoretical concerns raised are primarily due to a difference in convention between POMDP decision-making and generative modeling literature.
> We address each comment below and show that $\mathcal{I}$-VAE design choices are well posed given the structural properties of the POMDP:
>
> **(1)** We commit to improving the clarity of our generative model specification by refining our description of notation and by adding an appendix table defining each variable. We clarify that we here consider a special POMDP reduction in the EMDP with a static underlying state ($s$=$s'$) and not a traditional state-space model; therefore, p($z_t$ | $z_{t-1}$, $a_t$) does not apply. We further note that $z$ is an auxiliary latent variable for the generative model and not the hidden state. As the reviewer indicated, and to reiterate for clarity, $o$ is the observation, $s$ is the state, and $z$ is the latent.
>
> **(2)** The state encoder is used exclusively during training. At inference time, only the observation encoder path is active (mentioned in Sections 1, 3.1, and 3.2, and indicated in the CVAE Figure 2.b and I-VAE Figure 3). This is standard practice for conditional generative models, as seen in the original CVAE paper (Sohn et al., 2015). Note that we are using POMDP terminology for state $s$ to bridge generative modeling work with the belief updating research for POMDP problems. Same as our response for (1), we will clarify this in a table to ensure the nomenclature translation is clear.
>
> **(3)** We appreciate the Bayer reference, which we have reviewed carefully. Bayer’s core argument is that partial conditioning is harmful when omitted variables are not conditionally independent of $z$ given the included variables. To that end, Bayer provides safe cases in Section 3.3 in which partial conditioning is harmless:
>
> *“…the [conditioning] gap vanishes because $\bar{C_t}$ ⊥ $z_t$ | $C_{t}$ is approximately true”.*
>
> In our setting, this conditional independence is not only approximate; it is held exactly by the causal structure of the POMDP, as $z$ ⊥ $o$ | $s$ follows from d-separation. This is evidenced from the Bayes net for these models. The lack of performance degradation in Section 7 demonstrates this empirically, for which we provide a corresponding analysis. We further contend that $\mathcal{I}$-VAE without TCL matched CVAE performance on MNIST, suggesting this conditional independence is justified in practice.
>
> **(4)** The theoretical justification for TCL (as outlined in Eqs. 15-18) is distinct from how VAE loss optimizes a single-observation ELBO. TCL ensures learned latent representations are consistent across observation sequences from the same state. By contrast, while ELBO is proper for single-observation reconstruction, planning requires temporal consistency across observations that ELBO does not account for. This is posed as a complementary objective. As stated in Section 3.2, the TCL objective is not a heuristic; it optimizes a lower bound on mutual information via InfoNCE (van den Oord et al., 2018) which has well-established theoretical foundations.

---

### Review · Reviewer_RnrK · 2026-05-06

**Summary Of Contributions:**

The paper targets information-gathering POMDPs (specifically the static-state EMDP subclass) and makes three contributions. The first is the Inversion VAE (I-VAE), a CVAE variant that drops the observation from the recognition model and replaces the unit-Gaussian prior with a learned observation-conditioned prior, aligning the train- and inference-time use of the latent space. The second is a Trajectory Contrastive Loss (TCL) fine-tuning step that uses an InfoNCE-style objective to improve latent consistency across observations of the same state, helping in low-coverage regimes. The third is a Batched (Belief-State) MDP framework for GPU-parallel rollouts, supported by two theorems on value decomposition and on the equivalence between expected entropy reduction and expected posterior-to-prior KL divergence. The approach is evaluated on MNIST patch observations and on a real muon tomography intrusion discovery task.

### Strengths
* Clean, well-motivated I-VAE design that closes the train/inference gap of standard CVAEs.
* Clear gains from TCL fine-tuning in the sparse-observation regime.
* Valuable real-world muon tomography benchmark.
* Clear writing, figures, and overall presentations.

### Weaknesses:
* The empirical comparison should at least contain some other baseline models mentioned in the introduction section, other than the naive CVAE.
* Fig. 9 (table 1) and 14 show non-significant differences.
* Neither evaluation task has a known ground-truth posterior, so the quality of the learned posterior is only assessed indirectly through downstream task performance. A controlled synthetic experiment with a tractable true posterior would more directly verify that I-VAE approximates the true posterior rather than merely producing useful samples.

**Audience:**

Yes

**Audience Explanation:**

Belief-state planning, simulation-based inference, and ML for geosciences all overlap with the paper. The muon tomography task is an interesting experiment.

**Broader Impact Concerns:**

/

**Claims And Evidence:**

Yes

**Claims Explanation:**

But the significance of the experimental results (across several random seeds) is a weak point, as mentioned in the above box. And to make the proposed method more competitive, more baselines are necessary.

**Requested Changes:**

* Add a modern likelihood-free baseline (e.g., conditional score / diffusion posterior).
* Add a controlled synthetic experiment with a tractable true posterior to verify that the learned posterior is accurate, not just useful for the downstream task.
* Statistical reliability of the muon results. Report mean ± std over multiple training seeds and provide a paired test for the I-VAE vs. I-VAE-no-TCL and I-VAE vs. CVAE comparisons. Or, explain in detail regarding the significance level shown in those figures and their implications.

---

> ### Author Response · Authors · 2026-05-28
> **Rebuttal by Authors**
>
> We are grateful to the reviewer for their constructive feedback, and for identifying the $\mathcal{I}$-VAE design, TCL gains, and the muon tomography benchmark as strengths.
>
> **(1)** We agree to add a controlled synthetic inverse problem with a tractable true posterior, alongside a conditional diffusion posterior baseline. **More concretely, we introduce a four-component Gaussian-mixture inverse problem with noisy observations and a closed-form posterior. We compare $\mathcal{I}$-VAE, $\mathcal{I}$-VAE (no TCL), CVAE, and a conditional diffusion baseline against the analytical posterior using sliced Wasserstein distance (SWD), and report inference time for each. Diffusion sampling incurs a substantial inference-time cost: Diffusion-5 is approximately $9.5\times$ slower than $\mathcal{I}$-VAE, and Diffusion-10 is approximately $19.1\times$ slower. Despite this, $\mathcal{I}$-VAE achieves the lowest SWD across selected observation coverage levels. Benchmark results suggest that $\mathcal{I}$-VAE produces posterior samples that are both computationally efficient and closely aligned with the analytical posterior and supports our earlier statement that "iterative diffusion sampling can be substantially more expensive than a single-pass conditional generator".**
>
> **(2)** We acknowledge that the CLL confidence band overlap in Figures 9 and 14 is more pronounced at higher observation coverage. The TCL benefit concentrates in the low-coverage regime (below ~30% in MNIST, below ~25 observations in the muon problem), where separation between the methods is visible in the figures. This is precisely the regime most relevant to planning, where early decisions must be made with sparse observations. The operationally meaningful impact is visible in Table 2, where the $\mathcal{I}$-VAE + heuristic reaches a decision in 6.88 actions versus 20.65 for the CVAE + heuristic, a 3x improvement that reflects how small CLL gains compound through sequential action selection. We remove the bolding from Table 1, which initially reflected best performance rather than statistical significance, to minimize confusion, and add the above acknowledgement on variance reporting to the corresponding discussion.

---

### Review · Reviewer_FTw1 · 2026-05-13

**Summary Of Contributions:**

## Summary

The paper aims to address a setting described by so-called purely epistemic Markov decision processes (EMDPs), in which sequential decisions must be made based on histories of observations of a fixed but not directly observable state. The paper aims to address scalability. To scale up planning, the paper proposes the batched (belief-state) MDP: a dedicated cosmetic model to capture batching the simulation of states or beliefs. Since (1) batching of simulation benefits from parallelization and (2) observation likelihoods are not always available for the setting, the paper also proposes the inversion variational auto-encoder ($\mathcal{I}$-VAE): a dedicated conditional generative model to learn the density over states conditioned on the history of observations, equipped with a dedicated trajectory contrastive loss (TCL). Experiments are conducted on a synthetic benchmark based on MNIST and an intrusion discovery task using real-world tomography data.

## Strengths

In general, parts of the paper's exposition are well done and convincing.
- The proposed problem setting and the muon-based intrusion discovery benchmark are interesting and warrant more investigation, capturing real-world problems such as geological intrusion discovery with applications in automated mining. The muon-based intrusion discovery benchmark is a genuinely interesting geophysical sensing problem in which an explicit likelihood model is unavailable, and it is a well-motivated use case that warrants a generative modeling approach.
- The $\mathcal{I}$-VAE is well-motivated and described as a dedicated extension of the well-known (C)VAE models. Decoupling the recognition model from the observations via the conditional independence assumption and using a contrastive loss across observation sequences appears sound and well aligned with the self-supervised learning literature.
- The visual explanations throughout the paper are generally helpful and provide intuition for the model architecture. Additionally, it is important to note that code to reproduce experiments is provided.

## Weaknesses

The primary recommendation is to restructure the paper, with the $\mathcal{I}$-VAE as the central contribution, grounded in a formal EMDP problem statement, and to either substantially reduce the BB-MDP material or augment it with non-trivial analysis, see the weaknesses and recommendations below.
- MDPs, (information gathering) POMDPs, and epistemic MDPs (EMDPs) are not aptly introduced. While the intro provides a glimpse of EMDPs, the paper proceeds to the background of (C)VAEs and immediately introduces the $\mathcal{I}$-VAE in Section 3. The relevance of this material and the $\mathcal{I}$-VAE is hard to follow, as the paper poses no problem statement nor an (explicit/formal) introduction of the problem setting apart from an informal paragraph in the introduction, which is insufficient. The main problem setting, namely the EMDP and the inversion problem, its components, and its associated objective and optimal policy, is never formally introduced.
- While the computational advantages of batched simulations are clear, as evidenced by better inference time scaling, the motivation behind a dedicated "batched (belief-state) MDP" model is unclear. It is a well-established practice to parallelize simulations, often used for Monte Carlo (tree search) approaches. Sections 4, 5, and 6 consist mostly of the observation that optimality is preserved by batching multiple completely independent models and averaging their outcomes, which is not particularly surprising due to the linearity of expectations. There is insufficient motivation for this investigation and the resulting formalization. The paper should explain what the framework amounts to, even if it is just an elaborate notation for batching to enable computation speedups.
- Clarity: The signature of transition dynamics is deterministic functions, yet they are treated and described as stochastic. States are treated as discrete (with a sum) and continuous (with an integral) from one point to the next, while the actual state spaces tackled in the paper are never defined. Statements of Theorems 1 and 2 are deferred to the appendix, while they are referred to consistently in the paper. If these theorems are important to the paper's results, they should be in the main body. If not, then the results following the theorems should also not be in the main body.
- Statistical significance is not discussed in the results. Results that are slightly higher but remain within statistical confidence of each other are boldfaced. For example, in Table 2, the accuracy results for $\mathcal{I}$-VAE are boldfaced throughout, whereas for most entries, the reported error overlaps with that of $\mathcal{I}$-VAE (no TCL). The claim that TCL provides a meaningful improvement over just the $\mathcal{I}$-VAE rests partly on these results.

**Audience:**

Yes

**Audience Explanation:**

The proposed architecture may be of interest to those studying the use of generative models in EMDPs and applications in mining exploration in general. Due to the focus on the batched simulations, which aren't as interesting, the current state of the paper does not convey this information as effectively as possible, and major revisions may help place the paper's main findings more precisely within a formal EMDP problem statement.

**Broader Impact Concerns:**

While there may be ethical concerns regarding the mining applications studied by the authors, the paper itself poses no such direct ethical implications.

**Claims And Evidence:**

No

**Claims Explanation:**

As it currently stands, the paper's main claims are convoluted. It would be beneficial to establish the $\mathcal{I}$-VAE as the main contribution and place more emphasis on it, rather than the BB-MDP framework.

Some open questions regarding the $\mathcal{I}$-VAE remain:
- MNIST results indicate that the inversion architecture does not improve over CVAEs, but rather that the TCL provides the main benefit.
- The relation between the inversion architecture and the EMDP problem. Explicitly formalizing the EMDP and its associated problem statement in this paper will help ground the contribution of the $\mathcal{I}$-VAE.
- An ablation regarding the parameter $K$ in TCL seems warranted.
- A question arises on the action efficiency of the $\mathcal{I}$-VAE oracle/heuristic versus the CVAE oracle/heuristic: is this improvement driven primarily by the quality of the belief updater, or by some interaction between the belief updater and the entropy-based heuristic?

**Requested Changes:**

- Add a problem statement after a formal introduction of EMDPs: A dedicated section defining the EMDP, its components, its relation to the inversion problem, and the planning objective would address some of the clarity issues in the current framing of the paper.
- Cut back the sections on batched simulations in favor of grounding the $\mathcal{I}$-VAE as the main contribution. Alternatively, motivate and extend the (computational) advantages of the BB-MDP formalism more carefully to justify devoting such a large part of the paper to it.
- Fix notational inconsistencies: (1) stochastic vs deterministic models, (2) discrete (sums) vs continuous state spaces (integrals)
- Statistical reporting: results with overlap should not be bolded. At a minimum, the paper should report whether differences are statistically significant or note that they are not.

In my best judgment, all of the above are critical.

---

> ### Author Response · Authors · 2026-05-28
> **Rebuttal by Authors**
>
> We appreciate the reviewer’s careful and informed reading of our paper. We address the reviewer’s concerns regarding content focus, statistical significance, and notational inconsistencies, below.
>
> **(1)** We agree to the reviewer’s principal request to restructure the paper around the $\mathcal{I}$-VAE contribution. We have reframed the BB-MDP as a secondary contribution intended to supplement $\mathcal{I}$-VAE, and compressed BB-MDP content into a single section accordingly. Concretely, we moved discussion of batched value decomposition and optimality preservation, constructing the batched planning model, action selection, interpretation, and value function, to the technical appendix, while maintaining the BB-MDP definition and clarifying link to generative inversion methods for planning, which are parallelized by design, in the main text.
>
> **(2)** Additionally, we formally introduced the EMDP and a subsequent problem statement, preceded by only the minimal MDP/POMDP background needed to define the static-state information-gathering setting, following the introduction. We agree this will improve the paper's framing and ground the $\mathcal{I}$-VAE contribution more clearly.
>
> **(3)** Regarding the open question on $\mathcal{I}$-VAE (no TCL) vs CVAE performance: while the reviewer correctly notes that $\mathcal{I}$-VAE (no TCL) does not meaningfully improve over CVAE in MNIST as seen in Fig. 9, the performance difference is much more pronounced on the muon dataset. This makes sense; MNIST observations are direct masks of the underlying state, while muon observations are indirect, meaning that the learned conditional prior provides real benefit. We have adjusted bolding and more clearly stated statistical significance for Table 2.
>
> **(4)** The notational inconsistencies are a byproduct of introducing batched MDPs in Section 5 and batched belief-state MDPs (the belief space being a simplex) in Section 6. The MDP section uses standard discrete notation while the belief-state MDP section naturally involves integration. We agree that the logical transition from MDP to POMDP to belief MDP to epistemic MDP can be confusing to a general reader, especially since the paper focus is the latter. Similarly, epistemic MDPs maintain a static state, whereas the more general class of belief-state MDPs and POMDPs supported by the batched framework may involve stochastic state transitions. We mitigated these concerns through our new section formalizing the epistemic MDP and by consolidating the BB-MDP material into a single section using integral notation.
>
> **(5)** We move Theorem 1 and associated discussion to the appendix, with minimal reference in the main body. For Theorem 2, because it directly motivates the heuristic used in the experiments, we retain a concise statement in the main text and move the proof to the appendix.
>
> **(6)** On action efficiency: Table 2 suggests both effects. $\mathcal{I}$-VAE appears stronger than CVAE in settings that are not tied to the heuristic, especially under random and oracle policies. However, the performance difference is amplified when paired with the heuristic, where $\mathcal{I}$-VAE reaches a decision in 6.88 actions compared to 20.65 for CVAE. This suggests $\mathcal{I}$-VAE's better-calibrated uncertainty enables the entropy-based heuristic to select more informative actions.
>
> **(7)** We agree to conduct an ablation on the TCL parameter as requested **and do so on the MNIST benchmark. The results have been posted in the updated manuscript A.9, to include the newly introduced Figure 22. We find that TCL fine-tuning improves conditional log-likelihood over the no-TCL $\mathcal{I}$-VAE baseline across a broad range of $\alpha$, with the largest gains occurring in the 0-10\% low-observation regime. Ablation results indicate that the reported improvement is not sensitive to a narrow hyperparameter choice.**

---

> > ### Comment · Reviewer_FTw1 · 2026-06-03
> >
> > I thank the authors for their responses, and I look forward to the updated manuscript.

---

> > > ### Comment · Reviewer_FTw1 · 2026-06-09
> > > **Thanks for the changes: some remaining revision comments**
> > >
> > > I appreciate the author's efforts to restructure the paper in response to the reviews.
> > >
> > > Following the restructuring, there are some remaining comments, some of which I will expand on in more detail below.
> > >
> > > Primarily, I'm inclined to recommend putting the problem statement more prominently at the end of Section 2 and formalizing it. Reducing uncertainty is rather informal, I suppose the challenge is in balancing information gathering with minimizing sensing cost, which the POMDP framework addresses explicitly.
> > >
> > > Notation is a bit confusing at times:
> > > - Use of $O(\cdot \mid s,a)$ and $O(\cdot \mid a,s)$, and $o=f(s,a)$ for deterministic obsfun and $o \sim O(\cdot \mid a,s)$ for stochastic. The connection between POMDP beliefs $b(s)$ and the posterior $p(s \mid o_{1:t}, a_{1:t})$ should be made more explicit when introducing EMDPs.
> > > - $\mathbf{T}_b \colon \mathcal{B}^m \times \mathcal{A}^m \to \mathcal{B}^m $ is the signature of a _deterministic_ function. The steps in Eqs. 27-30 have a stochastic component due to $O$ being stochastic (in general). After Eq. (41), the authors write $\mathbf{b}' \sim \mathbf{T}_b(\cdot \mid  \mathbf{b}, \mathbf{a})$. I believe the signature must instead be $\mathbf{T}_b \colon \mathcal{B}^m \times \mathcal{A}^m \to \Delta(\mathcal{B}^m )$.
> > > - The combination of discrete and continuous state spaces remains very confusing. As far as I understand, the paper presents the Muon benchmark with continuous state spaces (though this is not very explicit). In Eq. (41), the entropy heuristic is defined with discrete sums over states. Confusingly, Theorem 2 in A.5 mixes discrete sums (e.g., Eq. 72) with integrals over the state space (e.g., Eq. 73).
> > >
> > > Some notes about the end of the paper in relation to the new changes:
> > > - The BB-MDP framework is a nice formalization of existing batching ideas for exploiting the parallelization capabilities of a neural network generative model, and now, thanks to the revision, it is positioned in the paper accordingly. The conclusion (in contrast to the abstract and the paper), however, starts with the BB-MDP as the main contribution.
> > > - Future works list investigating batched MCTS, or batched offline planning. The former is already a common practice and a field of study. For the second, it's unclear how a batched formulation speeds up value iteration over belief states. Do the authors intend to reference asynchronous value-iteration schemes? As it stands, such sentences do not carry enough meaning.
> > >
> > > Generally, given that the batched formalization is still a core part of the paper, I believe it justifies a literature review of prior work on batched/parallel MCTS, e.g., some works in chronological order:
> > >
> > > > Tristan Cazenave, Nicolas Jouandeau. A Parallel Monte-Carlo Tree Search Algorithm. Computers and Games 2008: 72-80
> > >
> > > > Guillaume Chaslot, Mark H. M. Winands, H. Jaap van den Herik. Parallel Monte-Carlo Tree Search. Computers and Games 2008: 60-71
> > >
> > > > Amine Bourki, Guillaume Chaslot, Matthieu Coulm, Vincent Danjean, Hassen Doghmen, Jean-Baptiste Hoock, Thomas Hérault, Arpad Rimmel, Fabien Teytaud, Olivier Teytaud, Paul Vayssière, Ziqin Yut: Scalability and Parallelization of Monte-Carlo Tree Search. Computers and Games 2010: 48-58
> > >
> > > > Shahaf S. Shperberg, Solomon Eyal Shimony, Ariel Felner: Monte-Carlo Tree Search using Batch Value of Perfect Information. UAI 2017
> > >
> > > > Panpan Cai, Yuanfu Luo, David Hsu, Wee Sun Lee (2021). HyP-DESPOT: A hybrid parallel algorithm for online planning under uncertainty. Int. J. Robotics Res. 40(2-3)
> > >
> > > > Marcus Hoerger, Muhammad Sudrajat, Hanna Kurniawati (2025). Vectorized Online POMDP Planning. CoRR abs/2510.27191
> > >
> > >
> > > And as cited by the authors:
> > > > Tristan Cazenave. Batch Monte Carlo Tree Search. In International Conference on Computers and Games,
> > > pp. 146–162. Springer, 2022.
> > >
> > > See also: `https://dblp.org/search?q=Parallel%20Monte-Carlo%20Tree%20Search`

---

> ### Author Response · Authors · 2026-06-16
> **Thank you for the added recommendations, and revision summary**
>
> We thank the reviewer for the detailed follow-up and for the suggestions that significantly helped improve manuscript clarity. We address the remaining comments and recommendations as follows:
>
> **(1)** We added a formal **"Likelihood-free planning in EMDPs"** problem statement, Problem 1, at the end of the new POMDP/EMDP background section. This problem statement defines the EMDP setting, decomposes the action space into sensing actions and terminal decisions, assigns sensing costs and terminal rewards, and states the planning objective. It also explicitly identifies the two central challenges addressed by the paper: likelihood-free belief updating and efficient evaluation of future observations and beliefs during planning.
>
> **(2)** We improved the observation-function notation throughout the manuscript. Deterministic observations are now written using the forward map $o=f(s,a)$, while stochastic observations are written using the observation kernel $o\sim O(\cdot\mid a,s)$. We also made explicit the connection between the POMDP belief and the posterior formulation in the EMDP setting by writing $b_t(s)=p(s\mid o_{1:t},a_{1:t})$.
>
> **(3)** We revised the stochastic signatures in the **BB**-MDP section. In particular, we write the batched belief transition as $T_b:B^m\times A^m\to\Delta(B^m)$, and similarly distinguish the vectorized state-transition and observation kernels, e.g., $T:S^m\times A^m\to\Delta(S^m)$ and $O:A^m\times S^m\to\Delta(\mathcal{O}^m)$. This clarifies that the vectorized update is deterministic only after the sampled observations are realized, while the induced **BB**-MDP transition is stochastic.
>
> **(4)** To address the concern about discrete and continuous state spaces, we revised the entropy heuristic and Theorem 2. We now distinguish the distribution-level expected entropy-reduction identity from the empirical entropy estimator used with particle beliefs. In the muon benchmark, the ground-truth state is described as a binary occupancy field, while the learned belief updater produces relaxed occupancy samples used to compute belief statistics. The former discrete state-sum heuristic has been replaced with a marginal particle-entropy proxy, and the appendix theorem is stated as an analytic information-gain identity. The notation table has been updated accordingly.
>
> **(5)** We added a related-work paragraph on parallel and batched online planning, including the suggested references on parallel MCTS, batch value of perfect information, HyP-DESPOT, and vectorized online POMDP planning. We now present **BB**-MDPs as complementary to existing parallel planning methods: our focus is on formalizing the batched belief-state transition induced by learned generative belief updaters in likelihood-free information-gathering POMDPs, rather than on replacing or rediscovering parallel tree-search methods.
>
> **(6)** Finally, we revised the conclusion to foreground the $\mathcal{I}$-VAE as the principal contribution, opening with the learned likelihood-free belief updater and then discussing the **BB**-MDP formalization as the planning framework that enables efficient batched use of these learned beliefs.
>
> The updated manuscript has been reattached for your continued consideration.

---

### Decision · Action_Editor_vdjF · 2026-07-06

**Recommendation:** Accept as is

**Audience:**

Yes

**Audience Explanation:**

The paper is of great interest to ML researchers working on planning and decision-making under partial observability.

**Claims And Evidence:**

Yes

**Claims Explanation:**

Overall, the paper is well motivated and demonstrates a compelling real-world application in muon tomography. However, the initial submission lacked clarity, and several claims were not sufficiently supported by the experimental evidence. The revision substantially improves the paper by providing a clearer problem statement, adding stronger baselines, and including additional ablation studies. Based on the latest version, all reviewers agree that the paper's main claims are now adequately supported by the evidence presented.